

# Mapping trends in woody cover throughout Namibian savanna with MODIS seasonal phenological metrics and field inventory data

Vladimir R. Wingate [1], Nikolaus J. Kuhn [1], Stuart R. Phinn [2], Cornelis van der Waal [3]

[1] Physical Geography and Environmental Change, University of Basel, Basel, Switzerland
[2] Remote Sensing Research Centre, School of Earth and Environmental Sciences, The University of Queensland, St Lucia, Australia
[3] Agri-Ecological Services, PO Box 28, Omaruru, Namibia

*Correspondence to*: Vladimir R. Wingate (Vladimir.wingate@unibas.ch)

**Abstract.** Woody vegetation is an integral component of savannas. Here, two main change processes alter woody vegetation, namely shrub encroachment and deforestation. Both impact a range of ecosystem services and functions across scales. Accurate estimates of change, including spatial extent, rate and drivers are lacking. This is primarily due to savanna vegetation
comprising woody and herbaceous vegetation, each of which exhibit divergent phenological characteristics, and vary importantly in their response to climatic and environmental factors. This study uses phenological metrics derived from the MODIS MOD13Q1 NDVI time-series to model woody cover as a function of field measurements, and to map trends across Namibia. These metrics enhance the contrasting phenological characteristics of woody and herbaceous vegetation, and standardizes their annual response to climatic and environmental factors by integrating short term variation. Trends in woody
cover are excellent indicators of shrub encroachment and deforestation. Trend significance was computed using the Mann-Kendall test, while change statistics, including the rate and spatial extent of change were derived using the Theil-Sen slope. Change was evaluated in relation to drivers including land-use, population, biomes and precipitation. An overall decrease in woody cover was identified, with the most pronounced decreases found in urban and densely populated areas. Decreases in woody cover were not homogenously distributed; losses predominated in tropical desert and dry forests, but gains were found
across shrub lands.

## 1   Introduction

Savannas constitute one of the most extensive biomes, covering over a fifth of Earth's land surface and providing a livelihood for a substantial number of people (Sankaran et al., 2005; Scholes and Archer, 1997). Savanna vegetation is characterized by the co-occurrence of trees, shrubs and herbaceous species with a distinguishing feature being their contrasting phenologies.
The proportions of woody and herbaceous species vary widely, forming forests, grasslands and shrub lands (Archibald and Scholes, 2007; Frost, 1996; Murphy and Bowman, 2012). Woody vegetation is an important component of savannas (Chidumayo and Gumbo, 2010; Sánchez-Azofeifa et al., 2005); it is essential not only at the local scale, where it provides resources for rural communities ranging from forage to timber, but also at the global scale, where it contributes key ecosystem services functions affecting biodiversity, carbon and water cycling, surface energy balance and climate (Adeel et al., 2005;





Alkama and Cescatti, 2016; Duveiller et al., 2018; Foley, 2005; Le Quéré et al., 2017; Turner et al., 2007). Woody vegetation is closely linked to biomass, carbon stocks, net primary productivity (NPP) and surface energy balance (Alkama and Cescatti, 2016; Foley, 2005; Le Quéré et al., 2017). Quantifying its dynamics is therefore an important global environmental issue and research priority (Poulter et al., 2014). For instance, it has been shown that both the trend and inter-annual variability of the

$CO_2$ uptake by terrestrial ecosystems are dominated by semi-arid ecosystems, of which savannas and woody vegetation are an important component (Ahlström et al., 2015). Moreover, recent research has shown that tropical dry forests, which are closely linked to savannas in sub-Saharan Africa, are far more extensive than previously thought, thereby highlighting the need for a greater understanding of ecosystem functions and services provided by these biomes (Bastin et al., 2017; Parr et al., 2014; Scholes and Archer, 1997).

Throughout sub-Saharan Africa, the need for arable land and forest products, such as charcoal, is driving widespread deforestation and forest degradation (Achard, 2002; Baccini et al., 2012; Brink and Eva, 2009). Simultaneously, a contrasting land degradation process is also widely reported, namely, the thickening of the woody layer and associated loss of herbaceous vegetation (i.e. shrub encroachment) which is of vital economic importance for rural communities relying on cattle (Liu et al., 2015; Ward, 2005). Both processes involve changes in woody vegetation cover and are often associated with pervasive land

degradation and desertification (Bond et al., 2010; Reynolds et al., 2007). At the same time, several authors questions whether shrub encroachment is a sign of declining ecosystem functioning, and suggesting that encroachment may limit degradation, boost carbon sequestration especially through the drylands (Eldridge et al., 2011; Maestre et al., 2009; Poulter et al., 2014; Soliveres and Eldridge, 2014). Thus, there is an inadequate understanding of the scale of woody vegetation change in relation to environmental and socio-economic and environmental drivers. In fact, separating trends in woody and herbaceous functional

groups is an active area of research (Brandt et al., 2016b; Helman et al., 2015; Roderick et al., 1999). As such, monitoring trends in woody cover and deciphering its relation to climatic and anthropogenic drivers, including precipitation and land management, respectively, is fundamental for sustainable land management and planning (Fensholt et al., 2012).

## 1.1   Satellite remote sensing

Satellite remote sensing offers a convenient tool to study trends in woody cover, due to its synoptic coverage and cost-

effectiveness once calibrated and validated. Several continental-scale tree cover products, available at the spatial resolution of Landsat and Moderate Resolution Imaging Spectroradiometer (MODIS), are widely used to monitor processes such as deforestation (Broich et al., 2014; Hansen et al., 2013a, 2016). These products use very high resolution (VHR) scenes to train a vegetation cover algorithm based on high to moderate resolution imagery. However, our current knowledge of the extent of tree cover and forests in drylands is limited. This is illustrated by substantial spatial disagreements between recent satellite-

based global forest maps (Hansen et al., 2013a; Sexton et al., 2013, 2016) and by the scarcity of large-scale studies of dryland biomes (Durant et al., 2012). Moreover, savannas exhibit pronounced land cover heterogeneity (Durant et al., 2012; Hansen et al., 2013b; Sexton et al., 2013, 2016), ranging from open grassland to closed forest while exhibiting high intra- and inter-




annual variability in vegetation photosynthetic activity and phenology, which is often enhanced by marked anthropogenic disturbances, including fire and grazing (Bastin et al., 2017; Gessner et al., 2013; Hansen et al., 2002, 2016).

Within this context, satellite-derived indicators of vegetation dynamics are fundamental to identifying environmental change processes such as land degradation. These indicators are often derived from spectral vegetation indices of satellite imagery,

which are related to the photosynthetic potential of vegetation canopies. For example, a time-series of Normalized Difference Vegetation Index (NDVI) effectively captures variation in photosynthetic activity, whether it results from phenological cycles or anthropogenic disturbances such as deforestation (Kuenzer et al., 2015; Myneni et al., 1995, 1997; Sankaran et al., 2008). The spatial and temporal resolution of MODIS is especially effective for continental-scale monitoring of land surface phenology, which refers to the seasonal and inter-annual variation in surface vegetation photosynthetic activity as measured

by satellite vegetation indices (Friedl et al., 2006). In addition, its high temporal imaging frequency allows the negative effects of cloud cover to be overcome; this is important in biomes which are seasonally impacted by cloud cover (Hansen et al., 2002; Jacquin et al., 2010). Regional-scale modelling of woody cover using MODIS is strengthened by including seasonal phenological metrics which describe contrasting stages in seasonal vegetation growing cycles (i.e. green-up and senescence), rather than simply metrics which represent temporal snapshots (Broich et al., 2014). Such metrics enhance the phenological

differences between woody and herbaceous vegetation, and constitute good proxies of woody cover, including all shrubs and trees, and have effectively been used to map trends in woody vegetation for large parts of sub-Saharan Africa (Horion et al., 2014). In addition, MODIS has successfully been used to monitor vegetation phenology (Zhang et al., 2003) and been used in conjunction with satellite-derived precipitation estimates (Zhang et al., 2005).

## 1.2    Phenology of Namibian savannas

In sub-Saharan savannas, important plant phenological events often occur in response to the seasonal availability of water, although it is widely reported that several vegetation phenophases do not follow this pattern. For instance, pre-rainy season leaf-flushing, as observed across much of sub-Africa, is thought to be driven mainly by photoperiod and temperature cues; here, woody species flower and leaf-out before the onset of the rains (Childes, 1988; Ryan et al., 2017). The many woody species found throughout Namibia exhibit different phenophases and vary in the timing of leafing and leaf senescence.

However, for most deciduous tree and shrub species, senescence occurs in response to a drop in soil moisture and the onset of lower daily minimum temperatures (Childes, 1988). The leaf-on period takes place shortly before the onset of the rainy season (September-October) and leaf-off period ensues throughout the middle of the dry season (April-August). Various species are evergreen, keeping at least a portion of their leaves throughout the dry season, but the majority are strongly deciduous, losing effectively all their leaves during the dry months and experiencing a leaf flush independently and before the first rains.  For

example, a pre-rainfall leaf flush and synchronized flowering is commonly observed in three tree/shrub species which are widespread in the northeast, in particular, *Terminalia sericea*, *Ochna pulchra* and *Pterocarpus angolensis*. In contrast, several common species in closed savanna woodland often demonstrate asynchronous flowering periods, notably *Baikiaea plurijuga* (Childes, 1988).  Trees and shrubs retain their foliage longer than herbaceous vegetation during the dry season (Mendelsohn





and el Obeid, 2005a; Verlinden and Laamanen, 2006). The annual growth of herbaceous biomass relies on the first precipitation events to initiate photosynthesis and remains photosynthetically active during the rainy season, as it is largely dependent on the spatio-temporal distribution of annual precipitation. Senescence of herbaceous vegetation then takes place at the onset of the dry season once the plants have completed their annual life cycle, while in addition, intense grazing pressure throughout

the country contributes to promptly grazing the pasture throughout much the country. Importantly, this results in woody vegetation remaining photosynthetic during part of the year, while herbaceous vegetation is entirely desiccated.

Woody vegetation dynamics in savanna biomes can be broadly divided into intra-annual (seasonal) fluctuations, and inter-annual trends (occurring over a period of years). Seasonal fluctuations are mainly controlled by seasonal precipitation and short-term anthropogenic disturbances, such as burning, grazing and vegetation clearing. Environmental factors such as soil

type, vegetation community composition and land-use history, also affect the seasonal growth of woody vegetation. Concurrent to shifts in canopy foliar density, are the associated leaf and plant phenology changes, including timing of flowering and leaf senescence; for most tree and shrub species, these are in turn regulated by seasonal water availability and temperature. For instance, the timing of plant phenological stages may vary for a given year, as a function of the seasonal availability of water in certain species, such as leaf flush occurring as a result of early rainfall onset. At the same time, these phenological stages

may be influenced by the climatic, anthropogenic and environmental factors listed above (Chidumayo, 2001; Childes, 1988; Kuenzer et al., 2015; Ryan et al., 2017; Wagenseil and Samimi, 2007). Importantly, these different processes, their interactions and synergies, contribute to creating a marked variation in the seasonal NDVI signal across savanna biomes. Finally, different plant functional types, including in particular herbaceous vegetation, also have a pronounced seasonal effect on the NDVI signal (Kuenzer et al., 2015).

In contrast, inter-annual trends impact woody vegetation dynamics over multiple years and thereby encompass the life cycle of trees and shrubs. For example, a minimum period of 5 years is thought sufficient to capture any increases in woody biomass, which is related to woody cover, using satellite indices (Asner et al., 2003; Ryan et al., 2012a; Williams et al., 2008). This is due to the fact that forest inventory parameters used to monitor tree growth, such as diameter at breast height, height, canopy cover and basal area, change slowly over the course of several years (i.e. gradual changes). However, they are also subject to

abrupt negative changes, often triggered by deforestation, encroacher bush control or forest fires (Chidumayo, 1997; Ryan et al., 2012b; Williams et al., 2008).

### 1.3   Aims

Against this background, land surface phenology across Namibia is highly variable over time, yet simultaneously reveals clear annual cycles at regional scales ($10^4$ km$^2$) (Wingate et al., 2018). It is to a large extent driven by the distinctive rainy season

(December-April) and the variable proportion of herbaceous and woody vegetation. These display divergent phenologies, which can be exploited to map either vegetation functional type (Hüttich et al., 2009; Mendelsohn and el Obeid, 2005a). In addition, vegetation change processes, including deforestation and woody encroachment are reported to be widespread in Namibia, yet their spatial and temporal dynamics remain little studied. In particular, the relation of these change processes to



land-use, biomes, population density and precipitation trends, are poorly known (Curtis and Mannheimer, 2005; John Mendelsohn and el Obeid, 2002a; Mendelsohn and el Obeid, 2005a; Wingate et al., 2016, 2018). The aims of the following study are therefore to 1) exploit the relationship between phenological metrics and field measurements of woody vegetation cover to create a time-series of percentage woody cover, 2) map trends in woody cover across Namibia, including the rates,

trajectory and spatial extent, and 3) evaluate the relationship between trends in woody cover and potential anthropogenic drivers, as well as environmental and climatic gradients, namely, land-use and population density, as well as biomes and precipitation, respectively.

## 2    Material and methods

### 2.1    Approach

An empirical/statistical approach based on the Random Forest algorithm was used to map percentage woody cover (Breiman, 2001; Colombo et al., 2003). Here, phenological metrics derived from a MODIS NDVI time-series, together with 484 field-based measurements of percent woody cover sampled over three years (2012, 2014 and 2016), were used to model percentage woody cover at annual intervals, resulting in a study period spanning the period from 2001 to 2016. Outliers NDVI values were attenuated by applying a temporal filter to the original MOD13Q1 NDVI time-series, and subsequently creating monthly

mean composites. Phenological metrics which integrate seasonal fluctuations into a single annual metric, characterising the phenology of a particular plant functional types were used to enhance long-term trends by suppressing inter-annual variation. The approach enables a maximum amount of noise and variation to be integrated, and hence enhances gradual trends and changes associated with woody vegetation. A fundamental underlying assumption of these metrics is that the observed NDVI signal during the dry season is derived only from woody vegetation. Thus, phenological metrics act as indicators of woody

vegetation cover, while also effectively separating the woody NDVI signal from the herbaceous one. Estimates were validated using independent measurements of tree cover, which are closely correlated with woody cover and hence constitute a proxy. Trend significance was computed using the Mann-Kendall test and the annual rate, trajectory and spatial extent of change was derived using the Theil-Sen slope. These were then evaluated in relation to potential drivers including land-use, population biomes and precipitation. Finally, hotspots of change were selected for further discussion (Figure 1).



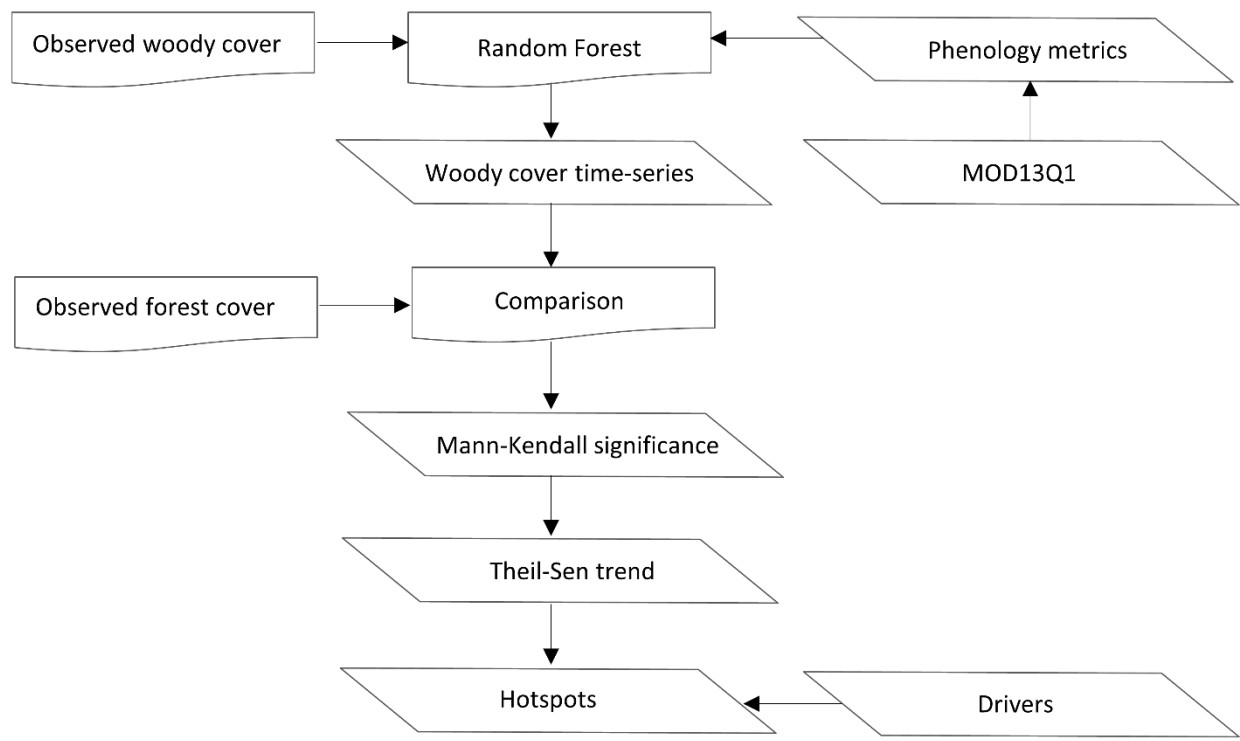

**Figure 1. Schematic workflow diagram illustrating the datasets and approach used to map and evaluate trends in percentage woody cover.**

## 2.2    Study region

The study area comprises the whole country of Namibia and encompasses an extensive and biogeographically diverse southern African savanna biome (Figure 2). The area is semi-arid to arid, with precipitation across the country varying from an annual average of 650 mm in the northeast, to 50 mm in the southwest. Rainfall events are variable both within and between years for any given period. In the north and central regions, precipitation is concentrated during the five summer months (December to April), while in the southernmost regions it occurs especially in the austral winter (Desmet and Cowling, 1999; Mendelsohn and el Obeid, 2005a). Agriculture ranges from subsistence small-scale cropping and ranching on commonages, predominantly across the northern areas, to large-scale commercial ranching and wildlife tourism enterprises on private (free-hold) lands in the central and southern areas; most of the land which is not set aside for conservation being used for livestock grazing or subsistence agro-pastoralism.



**Figure 2. The study area encompasses Namibia (822,634 km$^2$). The background image is a MODIS mean dry season NDVI image (2016), which enhances the presence of woody vegetation since herbaceous vegetation has already senesced.**



### 2.3    MODIS data

The MODIS MOD13Q1 NDVI time-series gridded level-3 product collection 6 has a number of advantages over the previous collection, making it suited for land surface phenology parameter estimation. A number of improvements have been made on the MODIS vegetation index algorithm, which minimize the confounding effect of bidirectional reflectance distribution functions (Didan, 2015). The product is available at 250m spatial resolution, is masked for water, clouds, heavy aerosols, cloud shadows and merged into 16-day composites, which allows only the highest quality values to be used, thus minimizing the impact of cloud cover (Didan, 2015). Pixels flagged as low quality were masked and a Savitzky-Golay smoothing filter was applied to each pixel of the time-series to interpolate missing values, smooth outliers and minimize the effects of low quality data resulting from noise and cloud cover, and the time-series was aggregated to mean monthly values. The computation of smoothing filters is recommended since observations over time are often noisy, especially in drylands (Broich et al., 2014).

### 2.4    Field data

Field measurements of percentage woody cover were made at three separate periods, 2012 (July), 2014 (May-June), and 2016 (April-May), across three separate regions and amalgamated into a single calibration dataset, resulting in a total of 483 samples. Only woody cover was used in this study since it aims to map vegetation associated with both deforestation and shrub encroachment, and woody cover is assumed to be representative of either plant functional type (i.e. trees and shrubs). Thus, woody vegetation encompasses both tall and short vegetation, counting trees and shrubs. Field data were collected using the point cover observation method described in Herrick *et al.* (2013), as part of the Land-Potential Knowledge System (Herrick et al., 2013), and consisting of stratified point intercept measurements of plant canopies in a $50 \times 50$ m area (Herrick et al., 2010). Sample sizes were 100 intercept points in 2012 and 2014, but were increased to 160 points in the 2016 survey. The post-processing and sampling effort was also different for the 2016 dataset, in which data were processed to fractional cover values. Therefore, in order for the 2012 and 2014 datasets to be included in the analysis, fractions of the three primary components (i.e. woody, herbaceous and bare ground) assessed were normalized so that their sum would correspond to 100 percent. In order to ensure the normal distribution, each variable was logarithmically transformed (Zandler et al., 2015). Samples with a measured percent woody cover <10% were excluded (n=25) from this analysis in order to apply log transformations, which otherwise would have resulted in negative values, this resulted in a total of 458 available for model calibration.

### 2.5    Scaling field data

Importantly, a key challenging in the use of field data in remote sensing research is making sure that the in situ field measurements provide a representative sample; this problem is especially pertinent in studies which cover large spatial areas with moderate spatial resolution data. In particular, when using plot-level field data to calibrate and validate remote sensing models at moderate spatial resolutions, the field data needs to be up-scaled to the resolution of the remotely sensed observations





(Baccini et al., 2007). In this study, the field plots were not scaled to the resolution of MODIS ($250 \times 250$ m) using spatial averaging; instead, the average percentage woody cover within the $50 \times 50$ m field plot was compared with the corresponding MODIS pixel values. We make the assumption, therefore, that the $50 \times 50$ m field plot adequately captures the average percentage woody cover within the corresponding MODIS pixel. We justify this assumption since the field plots were sampled 5 in homogenous vegetation strata (Baccini et al., 2007).

## 2.6 Spatial data

To investigate the observed trends in terms of potential drivers, a number of additional datasets were acquired. Land-use data were acquired from the Atlas of Namibia (Mendelsohn et al., 2002). Biomes distribution was downloaded from the Food and Agricultural Organization Global Forest Resources Assessment. For Namibia, they comprise tropical desert, tropical dry forest, 10 tropical mountain system and tropical shrub land; the latter two being very similar (Simons et al., 2001). Population density data were obtained from the Worldpop, high resolution global gridded dataset at 100 m resolution, which gives an estimation of the number of people per km² (Lloyd et al., 2017). This dataset was classified into four classes, with population densities per pixel ranging from 0-9, 9-53, 53-127, and 127-483 people per $100 \times 100$ m pixel. The average trends in percentage woody cover were then evaluated for each land-use type, biome and population density class.

15 ## 2.7 Rainfall data

Monthly precipitation was computed using the Climate Prediction Center Morphing technique (CMORPH) dataset, in which precipitation estimates are from satellite-derived passive microwave and infrared data, and available at a resolution of 0.0727° (Joyce et al., 2004). The CMORPH dataset was aggregated to mean annual values and converted to anomalies. To evaluate the correlation between rainfall and modelled woody cover, the CMORPH anomalies time-series was regressed, as the independent 20 variable, against the time series of annual percentage woody cover anomalies. In contrast to NDVI which is highly correlated with precipitation (i.e. NDVI integrates the signal from herbaceous vegetation which responds quickly to precipitation), we expect a low correlation between precipitation with predicted woody cover. This is because wooded regions undergo a pre-rainy season green-up (Childes, 1988; Fensholt et al., 2012; Herrmann et al., 2005; Nicholson et al., 1998; Nicholson and Entekhabi, 1987; Ryan et al., 2017).

25 ## 2.8 Modelling woody cover

The Random Forest algorithm was selected since it is effective at estimating predictor variable importance, integrating multiple predictor variables with different predictive power, not 'over-fitting' data, not assuming normal statistical data distribution or any particular relation (i.e. exponential) between dependent and independent variables, in addition to being multivariate (Breiman, 2001; Cutler et al., 2007; Moisen and Frescino, 2002; Prasad et al., 2006). 30 Models were created by taking plot measurements of percent woody cover, with the coincident pixel values of each of the five metrics (Table 1), for every year. In other words, the coincident pixel values of each of the five metrics do not correspond to





the timing of the field data; rather the field data, which have been aggregated into a single dataset, are related to each annual time-step of satellite data. This resulted in an annual time-series of modelled percentage woody cover for the period from 2001 to 2016.

Phenology metrics were extracted using TIMESAT, which has been extensively used for measuring seasonal land surface

phenology in drylands (Fensholt et al., 2012; Herrmann and Tappan, 2013; Horion et al., 2014; Jönsson and Eklundh, 2004). Three phenological metrics found to comprise an indirect measure of woody canopy cover were selected as model predictors (Table 1). They include the dry season index (DSI), described in Brandt et al. (2016) (Brandt et al., 2016a), the mean annual dry season values (MeanDS), and the dry season integral (DSINT) defined in Brandt et al. (2016) (Brandt et al., 2016b). These metrics reduce the confounding effects of herbaceous vegetation on woody vegetation, by taking advantage of the seasonal

period when herbaceous vegetation photosynthetic activity and biomass are at their minimum (i.e. the dry season).

In order to test the sensitivity of these metrics to predicting woody cover from field measurements, two phenological metrics known to be related to herbaceous vegetation cover were included in the model. Predictor variable importance metrics were then computed to contrast their effectiveness at predicting observed woody cover. They include, the maximum annual values (MaxWS) taken for each wet season, and the integral of the difference between the function describing the season and the base

level from season start to season end or the small seasonal integral (SINT) (Jönsson and Eklundh, 2004). Calculation of the SINT used values spanning the start of season (SOS) and end of season (EOS), as described in Jönsson and Eklundh et al. (2004) (Jönsson and Eklundh, 2004), where the onset of the rainy season is inferred by assigning a percentage threshold of 20% of the yearly NDVI amplitude value. Although this threshold is well established for the SOS, it should be interpreted with caution for the EOS, since a number of factors are likely to influence this period, for example, wetlands and cropland remaining

photosynthetically active longer than the surrounding herbaceous layer, or a particularly wet year may result in vegetation remaining photosynthetic for longer.

**Table 1. Phenological metrics used in this study, their abbreviation and concise description.**

| Phenological Metric | Short form | Description |
|---|---|---|
| Annual dry season index | DSI | Dry season index, calculated as per Brandt *et al.* (2016) |
| Mean annual dry season value | MeanDS | The mean annual values taken for the duration of the dry season |
| Maximum annual wet season value | MaxWS | The maximum annual values taken for each wet season |
| Annual small seasonal integral | SINT | Integral of the difference between the function describing the season and the base level from season start to season end |
| Annual dry season integral | DSINT | Dry season integral described in Brandt *et al.* (2016) |

## 2.9    Model accuracy and comparision

The paired observed and predicted values were used to compute two accuracy metrics, namely, the Root Mean Squared Error (RMSE) and the coefficient of determination ($R^2$) (Stehman et al., 2012; Willmott, 1982). For the Random Forest algorithm where the output is a continuous response, the pixel-wise prediction represents the average of the model trees. Hence, the



individual tree predictions offer the opportunity to compute uncertainty metrics including the coefficient of variation (CV) and standard deviation (SD); a large CV and SD suggests higher variability (Zhu and Southworth, 2013). Finally, model predictions were compared to the recently published 4,684 sample calibration/validation dataset of percentage tree cover from Bastin *et al*. (2017) (Bastin et al., 2017); each data point consists of a 0.5-ha plot, visually assessed for tree cover percentage using very

high resolution imagery. This dataset is assumed to act as a good proxy for percentage woody cover, since interpretation of contemporary high resolution imagery was used and it provides the latest estimate of tree cover in drylands. In addition, we find observed tree cover percentage and observed woody cover percentage, sampled as part of this study, to be highly correlated ($r$=0.83).

To test the correlation between modelled percentage woody cover and the validation dataset, model predictions were firstly

classified into 5% increment classes; the validation dataset was then aggregated and averaged within these classes. The mean values of the validation dataset were regressed with the corresponding values of 5% increment class values, and the $R^2$ and RMSE computed. This approach aimed to reduce the spread of values from the large number of validation points. No validation has been conducted in the temporal domain due to a lack of long-term field data which would have allowed validating past land cover changes.

**2.10   Trend analyses**

Key aspects surrounding trend estimation from Earth Observation (EO) time-series include temporal and spatial resolution, as well as data quality (Badreldin and Sanchez-Azofeifa, 2015; Sulkava et al., 2007). Although trend estimation using linear regression analysis is widely employed, it contravenes several statistical assumptions (deBeurs and Henebry, 2004; Eklundh and Olsson, 2003). Hence, non-parametric tests which overcome these limitations were applied (i.e. Mann-Kendall and Median

Theil Sen trend analyses) (deBeurs and Henebry, 2004; Forkel et al., 2013). Furthermore, limitations are incurred by temporally aggregating, for example, to the annual scale, by diminishing temporal resolution. On the other hand, annual aggregation may strengthen trend analysis by eliminating seasonal cycles, which have been found to add seasonal correlation structures and thus augmenting uncertainties (Forkel et al., 2013). In this study, by aggregating NDVI values to average monthly scales, we assume fluctuations due to climate, fire or anthropogenic activity are effectively integrated, permitting the quantification of

anomalies, such as deviations from long-term averages, and the strengthening of the signals under investigation, namely that of woody vegetation.

The time-series was first converted to anomalies before applying the trend analysis (Eastman, 2009). Subsequently, the statistical significance of the trend was defined by applying a pixel-wise Mann-Kendall trend test to the woody cover time-series. Areas which exhibited no significant trend (P≥0.05) were masked out and assumed to represent no change. For areas

which demonstrated a significant trend (P≤0.05), the Theil-Sen trend test was applied to the time series. This approach smooths the annual time-series using a linear trend, and is generally recommended for looking at rates of change in noisy or short time-series, since it is robust in identifying trends and insensitive to outliers (Hoaglin et al., 1983). Additionally, it has been



extensively used to measure trends in EO time-series (Andela et al., 2013; Fensholt et al., 2012; Guay et al., 2014; Zhu et al., 2016).

The percentage woody cover time-series was created with the aim of reducing both inter- and intra-annual fluctuations; however, important inter-annual fluctuations in the percentage woody cover time-series values remained apparent. These can
be attributed to a variety of factors, including variable atmospheric and weather conditions, together with changing vegetation phenology. In light of these fluctuations, simply applying an image differencing change detection method to estimate annual change would not be reliable; hence, the Theil-Sen trend slope was used. By multiplying the slope by the number of years, the annual rate of change rate can be computed, thereby providing an estimate of change in percentage woody cover between 2001 and 2016. The resulting annual slope was then spatially aggregated by taking the mean (spatially aggregated mean annual net
change) to the country level, land-use types, population density classes and biomes; the mean slope, minimum and maximum change estimates reported. Subsequently, two layers were derived from the significant annual slope image, the percentage woody cover gain (positive slope) and loss (negative slope). Pixel-wise loss and gain were then aggregated (mean values) at the level of the country and land-use types, to estimate gross gain and loss (i.e. loss and gain) (Table 2). All datasets were projected to the WGS_1984_UTM_Zone_33N projected coordinate system, and annual slope values we converted from the
MODIS resolution ($250 \times 250$ m) to km$^2$ ($1000 \times 1000$ m) by multiplying by and expansion factor (1.6), where:

Equation (1)

$$exapnsion\ factor = \frac{100,000m^2}{62,500m^2}$$

## 2.11 Multi-temporal imagery evaluation

To qualitatively assess what the observed trends represent on the ground, in terms of land cover change, a visual assessment was undertaken using a range of multi-temporal, high resolution scenes, including Google Earth imagery. The assessment aimed to identify direct drivers namely, human driven land-use and land cover change, such as urbanization and deforestation, as well as indirect drivers, including land cover changes driven by a combination of climate and human land-use, such as woody encroachment. Two classes were created representing areas mapped as either positive or negative trends, with slopes
$\geq 25\%$ ($\geq$-25). 10 randomly sampled points were then generated within each class and subjectively assessed for vegetation changes using Corona satellite imagery (6 feet spatial resolution, 1972), contemporary aerial images (0.5 m, 2010), pan-sharpened Landsat 7 imagery (~15 m, 2000), Sentinel-2 imagery (10 m, 2016) and multi-temporal (time-slider tool) Google Earth imagery.



## 3 Results

### 3.1 Predictor layer importance and model uncertainty

The evaluation of predictor importance yielded a clear pattern: the maximum annual wet season value (MaxWS) and annual small seasonal integral (SINT) were consistently the weakest predictors (expect for the 2007 model, in which the DSINT is

5    the weakest, potentially implying an anomalous year). Predictor importance (2008) is plotted in (Figure 3); two measures are used to assess variable importance, including percent increase in Mean Standard Error (MSE) following random permutation, and increase in node purity resulting from all the splits in the forest based on a particular variable, as computed using the gini criterion. Plots for the remaining years (2001-2015) are provided in the supplementary material (Figure S1). A plot of CV shows that model uncertainty is higher for the arid coastal and southern regions, where percentage woody cover lower and

10   most likely more variable; where percentage woody cover is high and more stable, a lower CV is identified (Figure 3). These results indicate that in areas of lower woody cover, model predictions are the least accurate and should be interpreted with caution.

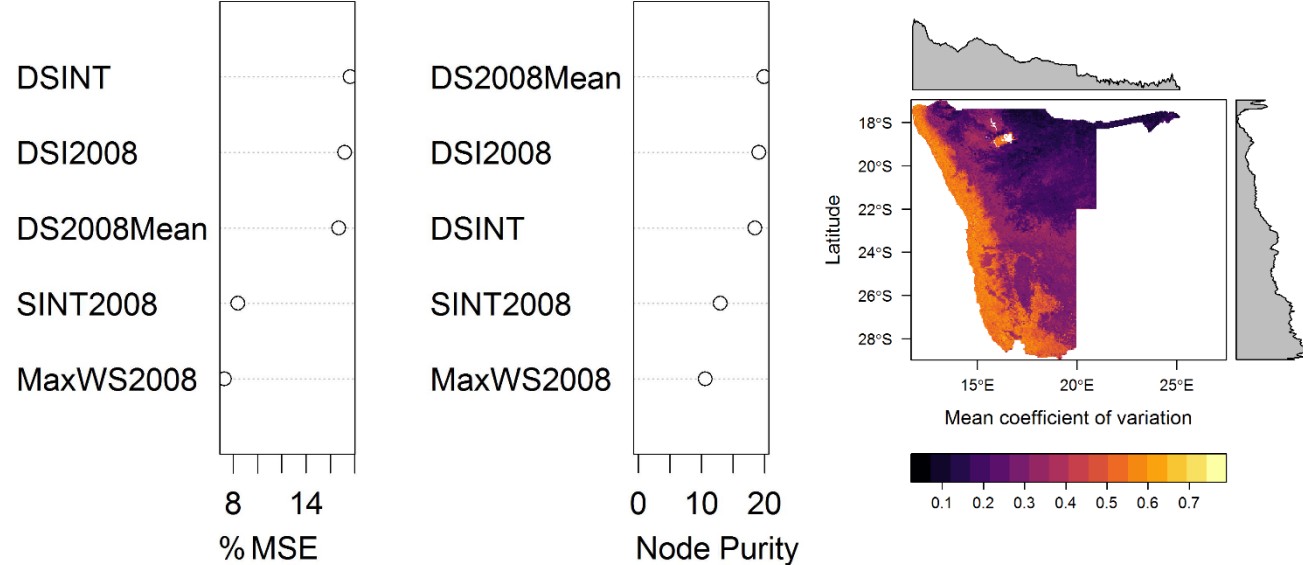

**Figure 3. Predictor importance (2008) generated using the Random Forest algorithm. Mean coefficient of variation.**

15   ### 3.2 Model accuracy and comparison

Moderate correlations were found between observed and predicted (2001-2016) percentage woody cover. Figure 4 illustrates the linear relation between observed and predicted (2016) percentage woody cover, yielding an $R^2$ of 0.47 and an RMSE of 14.47%. Between the 2001 and 2016 models, the $R^2$ values ranged from 0.4 to 0.5, and the RMSE ranged from 14.14% to 15.43%. Plots for the remaining years are provided in the supplementary material (Figure S2).



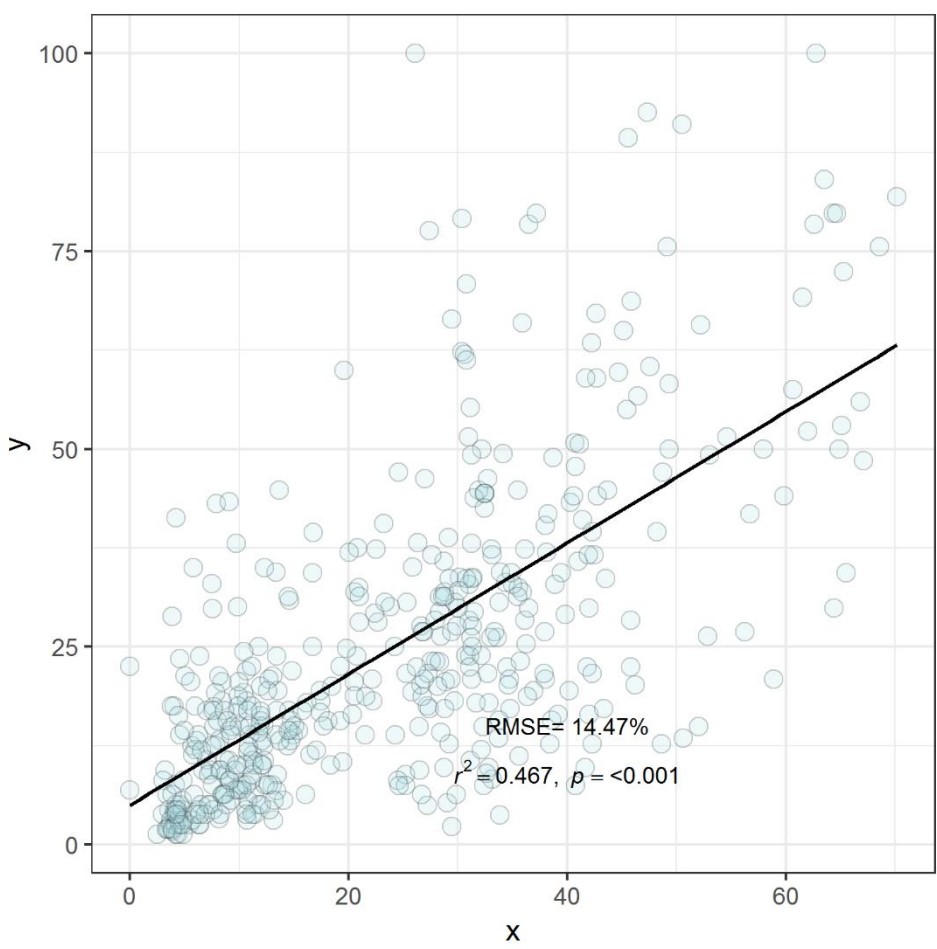

**Figure 4. Linear relation between observed and predicted (2016) percentage woody cover.**

Each annual model prediction was compared to the Bastin *et al.* (2017) percentage tree cover dataset (Bastin et al., 2017). Figure 4 illustrates the linear relationship between percentage woody cover at 5% increment classes (2016), and percentage tree cover, yielding an $R^2$ of 0.77 and an RMSE of 3.94%. Between the 2001 and 2016 models, the $R^2$ values ranged from 0.13 to 0.96, and the RMSE varied from 3.52% to 4.10%. Plots for the remaining years (2001-2015) are provided in the supplementary material (Figure S3).



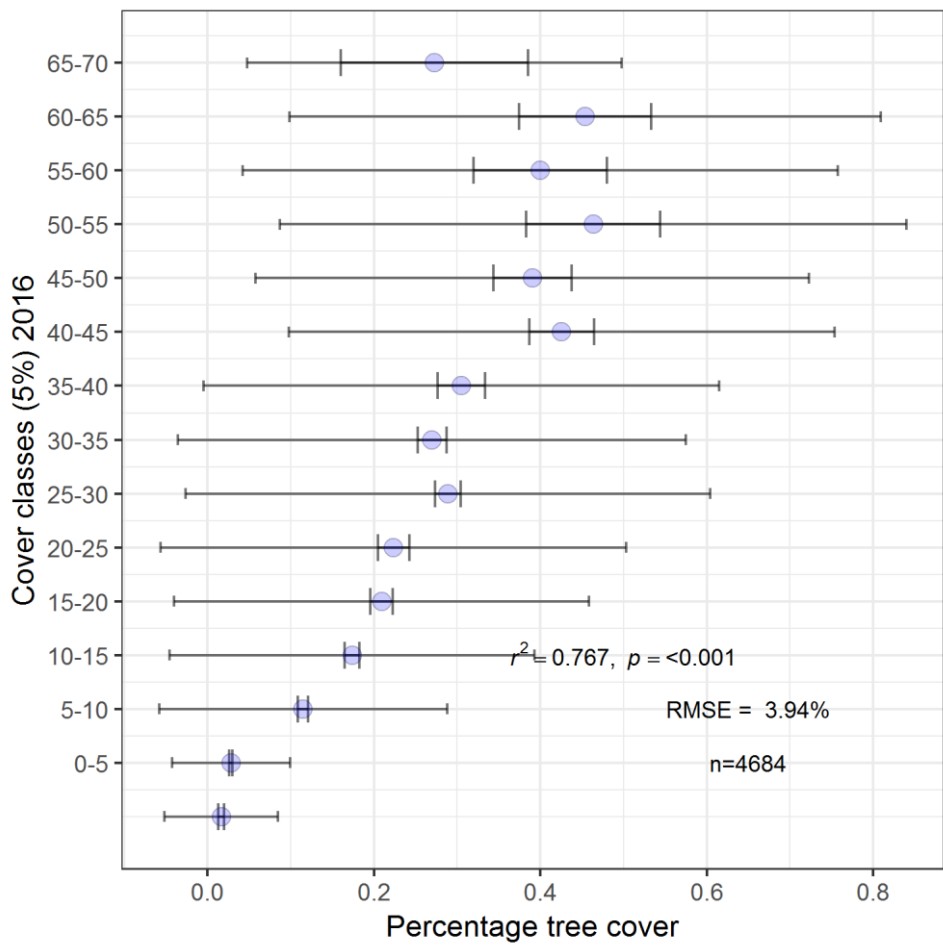

**Figure 5. Linear relationship between percentage woody cover at 5% increment classes, and percentage tree cover.**

### 3.3 Trends in relation to land-use, biomes and population

Trend analysis results for Namibia and its constituent land-use classes are presented in Table 2. The areal extent of modelled

5 woody cover in 2001 (km$^2$), the spatially aggregated mean annual net change (i.e. slope [% km$^2$ yr$^{-1}$] and minimum and maximum values [% km$^2$ yr$^{-1}$], and total change (i.e. loss and gain) (% km$^2$ yr$^{-1}$) are shown. The average annual slope for Namibia is negative (-4.38 % km$^2$ yr$^{-1}$). In term of land-use types, large-scale communal and urban lands show a positive average annual slope with 8.42 % km$^2$ yr$^{-1}$, and 0.37 % km$^2$ yr$^{-1}$, respectively. For the remaining land cover types, mean annual slope was negative, suggesting an overall loss of woody cover (Table 2).



**Table 2. Change in woody cover (annual slope) in relation to land-use, estimated using Theil–Sen trend test, of the time series of annual percentage woody cover area. The 2001 percentage woody cover area, Min and max slopes are included. P represented a Mann–Kendall trend test with P < 0.05 used to define statistically significant trends, with a sample size of n = 16 years. Total change in percentage woody cover was estimated pixel-wise using the Theil-Sen trend test, with losses and gains being summed and converted to km² to compute total loss and gain.**

| Percent woody cover area (2001) | | | | | | | |
|---|---|---|---|---|---|---|---|
| | | Mean annual net change | | | | Total change | |
| Land-use | Woody cover/km² | Slope/ km² yr⁻¹ | Min/ km² yr⁻¹ | Max/ km² yr⁻¹ | | Loss km² yr⁻¹ | Gain km² yr⁻¹ |
| Namibia | 774588 | -4.38 | -4.82 | 5.42 | | -303 | 257 |
| Large-scale communal | 354332 | 8.42 | -3.55 | 4.05 | | -21 | 20 |
| Resettlement | 5257 | -12.59 | -3.17 | 2.42 | | -21 | 12 |
| Government agriculture | 48697 | -8.11 | -2.58 | 3.65 | | -20 | 27 |
| Other government | 16159 | -12.32 | -3.07 | 2.72 | | -18 | 6 |
| Urban | 6741 | 0.37 | -4.67 | 5.42 | | -18 | 21 |
| State protected | 246907 | -6.26 | -3.58 | 2.98 | | -18 | 15 |
| Small-scale Communal | 88852 | -14.94 | -3.57 | 3.87 | | -19 | 16 |
| Freehold | 6982 | -7.76 | -4.82 | 4.14 | | -20 | 11 |







**Figure 6. Maps the significant positive (negative) Theil-Sen trend slope, including hotspots of change selected for further discussion marked in black rectangles (c-h), overlaid on modelled percentage woody cover for the study area in 2016. Positive trends are shown in blue and negative in red.**

Tropical shrub land manifested very minor decline (-0.17 % $km^2$ $yr^{-1}$) in woody cover and associated tropical mountain system

an increase (3.76 % $km^2$ $yr^{-1}$), respectively. The tropical desert biome displayed a pronounced decrease (-4.64 % $km^2$ $yr^{-1}$) in

woody cover, while the tropical dry forest biome experienced the most striking decline (-7.39 % $km^2$ $yr^{-1}$) (Table 3).

**Table 3. change in woody cover (annual slope) in relation to biomes.**

| Percent woody cover area (2001) | | | | | | | |
|---|---|---|---|---|---|---|---|
| | | Mean annual net change | | | | Total change | |
| Biome | Woody cover/$km^2$ | Slope/ $km^2$ $yr^{-1}$ | Min/ $km^2$ $yr^{-1}$ | Max/ $km^2$ $yr^{-1}$ | | Loss $km^2$ $yr^{-1}$ | Gain $km^2$ $yr^{-1}$ |
| Tropical desert | 2698569 | -4.64 | -31.96 | 36.68 | | -8.70 | 1.76 |
| Tropical mountain system | 182080 | 3.76 | -25.23 | 27.43 | | -8.34 | 7.59 |
| Tropical shrubland | 3421476 | -0.17 | -48.24 | 37.50 | | -11.27 | 9.58 |
| Tropical dry forest | 1432130 | -7.39 | -46.78 | 54.29 | | -12.79 | 15.31 |

Most of the country (>99.86%) has a low population density, with between 0-9 people per $km^2$ (Table 4). Areas of "no data"

cover a relatively large area (0.11%), while the remaining population density classes (0-9, 9-53, 53-127, 127-483 people per

$km^2$) occupy 0.03% of the remaining land area. The average slope values for each population density class are listed in Table

4. All population density classes show an average negative trend, with the strongest decline being in the middle 9-53, 53-127

classes, (-11.10 % $km^2$ $yr^{-1}$, -8.12 % $km^2$ $yr^{-1}$, respectively).

**Table 4. Change in woody cover (annual slope) in relation to population density classes.**

| Percent woody cover area (2001) | | | | | | | |
|---|---|---|---|---|---|---|---|
| | | Mean annual net change | | | | Mean annual change | |
| Population density | Woody cover/$km^2$ | Slope/ $km^2$ $yr^{-1}$ | Min/ $km^2$ $yr^{-1}$ | Max/ $km^2$ $yr^{-1}$ | | Loss/ $km^2$ $yr^{-1}$ | Gain/ $km^2$ $yr^{-1}$ |
| NoData | 573.42 | -3.43 | -50.40 | 69.31 | | -22.23 | 24.08 |
| 0-9 | 773104.77 | -4.38 | -77.18 | 86.86 | | -18.93 | 16.08 |
| 9-53 | 223.32 | -11.10 | -48.55 | 7.60 | | -13.61 | 3.95 |
| 53-127 | 20.65 | -8.12 | -23.98 | -3.52 | | -8.12 | 0.00 |
| 127-483 | 6.74 | -4.20 | -4.20 | -4.20 | | -4.20 | 0.00 |

### 3.4    Trend assessment using multi-temporal imagery

Of the 20 randomly sampled points assessed for each trend slope class (i.e. ≥25% and ≥-25%), two examples exhibiting

characteristic land cover changes were selected for further discussion. They include a direct human impact, namely land

clearing (shown in Figures 6a and 6b), and an indirect impact, potentially manifesting as greening (shown in Figures 7a and

7b). For the remaining randomly sampled points in the ≥25% trend class, no clearly apparent land cover changes could be

distinguished, using the available imagery and sampling method.









**Figure 7. Randomly sampled point for an area exhibiting a significant negative slope (≥-25%), visible as land clearing for small-scale agriculture and indicative of direct land cover change. These are identified using a 1972 Corona image (a) and a 2010 aerial othrophoto (b).**







**Figure 8. Randomly sampled point for an area exhibiting a significant positive slope(≥25%); no apparent change can be identified from a 1972 Corona image (a) and a 2010 aerial othrophoto (b). Results may be indicative of indirect change.**

### 3.5    Trends in relation to precipitation

The linear regression between mean annual precipitation anomalies (independent) and annual percentage woody cover anomalies (dependent), reveals that the majority of $R^2$ values are low, signifying no significant linear relationship. The result implies that anomalies in precipitation are not coupled with those of percentage woody cover for most of the country, except along the western escarpment (Figure S4).

## 4    Discussion

### 4.1    Trends in relation to biomes

Globally, arid and semi-arid (desert) biomes have recently been found to exhibit large decreases in short vegetation (≤5 m in height) and important increases in bare ground, with both trends pointing to long-term land degradation; simultaneously, the world's tropical shrub land biome is reported to have experienced a considerable areal increase in short vegetation and a concurrent bare ground loss, and these results are postulated to be the result of woody encroachment (Song et al., 2018). In contrast, the tropical dry forest biome is found to have undergone significant levels of deforestation (Song et al., 2018). When evaluating the results from Song et al. (2018) for Namibia only, an overall greening trend from 1982 to 2016 can be identified. On average for each FAO biome, a decrease in bare ground and a simultaneous increase and short vegetation can be noted, while a gain in tree canopy is seen across the tropical dry forest biome. Our results differ in that we identify an overall browning trend with an average slope of -4.38 km$^2$ yr$^{-1}$, with an especially marked decrease in woody cover across the tropical dry forest biome (-7.39 km$^2$ yr$^{-1}$). These contrasting results may be due to the different spatial (0.05° × 0.05° compared to 250 m × 250 m) and temporal (1982-2016 compared to 2001-2017) scales of the studies.  Importantly, they serve to highlight how these two factors can lead to substantially differing results when analysing EO time-series.

Of the four biomes which Namibia encompasses, large parts of the country are desert (38.45%), shrub land (including mountain system) (43.74%) and dry forest (17.81%) (Simons et al., 2001). Tropical shrub land showed only a very small decline in woody cover (-0.17 % km$^2$ yr$^{-1}$), and the closely related tropical mountain system an increase (3.76 % km$^2$ yr$^{-1}$), indicating overall woody encroachment. In contrast, the tropical desert biome showed a marked decline (-4.64 % km$^2$ yr$^{-1}$) in woody cover, suggesting long-term land degradation. Lastly, the tropical dry forest biome experienced the most pronounced decline in woody cover (-7.39 % km$^2$ yr$^{-1}$), pointing to extensive deforestation (Table 3).

Overall, our results point to the occurrence of contrasting land cover change processes, with both gains and loss in woody cover. We find that woody cover loss is associated with the more humid areas (tropical dry forest) and is therefore potentially the result of deforestation/forest degradation , which has been shown to be taking place (Wingate et al., 2016, 2018).  The desert biome also exhibited a woody cover loss; such losses in arid biomes are often associated with desertification and land





degradation (Song et al. 2018b). In contrast, the tropical shrub land and mountain biomes demonstrated an increase woody cover suggesting greening and in turn shrub encroachment (Saha et al., 2015). Our results mirror those recently publish by Brandt et al. (2017); in their pan-African study on trends in woody cover, the authors found that negative trends were preferentially associated with humid, high biomass forest biomes, while positive trends were mostly found across drylands,

except very hot xeric ecoregions or tropical deserts (Brandt et al., 2017).

## 4.2    Trends in relation to land-use and population

The overall positive trend in percentage woody cover (8.42 % $km^2$ $yr^{-1}$) identified for large-scale agriculture on communal land agrees with previous studies, in that a high density of encroacher species, at early growing stages, were identified in these regions (De Klerk, 2004a). However, they may also result from other land management practices, such as agro-forestry and

large-scale fencing of commonage, which are known to cause an overall increase in woody and herbaceous vegetation (John Mendelsohn and el Obeid, 2002b). However, this result is also unexpected, since the land-use type has been shown to be experiencing substantial land clearing for cropping and ranching (Wingate et al., 2016, 2018). The limited increasing trend (0.37 % $km^2$ $yr^{-1}$) observed for urban land areas is again unexpected, since rapid urbanization, associated with vegetation cover losses, is occurring throughout Namibia (Wingate et al., 2016).

A decrease of -14.94 % $km^2$ $yr^{-1}$ in woody cover is identified across small-scale agriculture on communal land; while a similar loss of -12.59 % $km^2$ $yr^{-1}$ can be noted on resettlement land. This trend is most likely the result of widespread vegetation clearing for small-scale cropping and ranching (John Mendelsohn and el Obeid, 2002a; Wingate et al., 2016). Importantly, it is likely that as a consequence of the moderate spatial resolution of MODIS, much of the small-scale deforestation is being concealed; in fact its resolution has been shown to hide up to 50% of small-scale deforestation (Anderson et al., 2005; Hammer

et al., 2014; Hansen and Loveland, 2012). The negative trend identified across protected areas (-6.26 % $km^2$ $yr^{-1}$) is unexpected, since conservation efforts in Namibia, especially in the Kalahari woodland ecoregion, focus on the preservation of woodlands and forests (Mendelsohn and el Obeid, 2005b). However, long-term fire-scar monitoring studies throughout the northern regions of the country, where much of the conservation areas are found, identify increasing fire frequencies. More frequent fires are associated with a decrease in tree stem diameters, densities and species diversity, and is potentially being driven by

more intensive land management (Anon, 2017; De Cauwer et al., 2016; Mendelsohn and el Obeid, 2005b; Sankaran et al., 2008) (Table 2).

An important environmental and socio-economic question for much of northern Namibia and neighboring countries is the expansion of small-scale arable cropping into marginal land (Pröpper et al., 2010). The primary reason for this expansion is the demand for farm land due to population growth; in addition, the on-going land reform is introducing land privatization and

hence important changes in land-use on commonages, for instance, large-scale fencing (Mendelsohn and el Obeid, 2005b). Furthermore, since the end of the civil war in 1990, the region has undergone important infrastructural developments, with new roads connecting much of the north and neighboring countries, together with the establishment of water and power infrastructure. These have greatly facilitated access to and settlement of remote regions, promoting the expansion of new




farmsteads and villages. The greater need for arable land and the consequent heightened pressure on it for the extraction of natural resources such as timber, lead to a reduction in woody cover (De Cauwer et al., 2016; Pröpper et al., 2010; Röder et al., 2015; Schneibel et al., 2013, 2017; Wingate et al., 2016).

Across agriculture and tourism on freehold land, the negative trends observed (-7.76 % km$^2$ yr$^{-1}$) may be the result of
encroacher shrub control (De Klerk, 2004a). This land management activity is widely implemented on commercial farm land in order to favor herbaceous vegetation growth which supports livestock production (Mendelsohn and el Obeid, 2005c) (Table 2).

The 9-53, 53-127 population density classes exhibited the largest declines in woody cover (-11.10 % km$^2$ yr$^{-1}$, -8.12 % km$^2$ yr$^{-1}$, respectively), suggesting densely populated regions are responsible for most land cover changes associated with decreases
in woody vegetation cover. Table 4 shows the mean trend values for each population density class as being largely negative, indicating that overall, decreases in woody cover occur across population density classes, and implying that population density may have a role in explaining the observed trends.

### 4.3 Trend assessment using multi-temporal imagery

Qualitative analysis of historical and contemporary high resolution imagery, using a random sample of ten points distributed
in areas with slopes ≥25% (≤-25%), enabled the observed trends to be interpreted as specific land cover changes. In particular, negative trends were chiefly identified as a general reduction in vegetation cover, together with a concurrent increase in bare ground cover, vehicle tracks and farm plots, thereby indicating a direct human impact (Figure 7). However, when evaluating areas exhibiting positive trends, land cover changes were harder to conclusively identify. For instance, no apparent change can be seen in Figures 8a and 8b, yet this area was mapped as having undergone a significant positive trend. These results suggest
the occurrence of indirect impacts manifesting gradually; as such they may comprise increases in vegetation density.

### 4.4 Trends in relation to precipitation

Low $R^2$ values, resulting from the linear regression between percentage woody vegetation cover anomalies and precipitation anomalies, are seen across much of the country. The of cause of this may be that much of the country's densely wooded areas are photosynthetically active before the on-set of the rainy season i.e. pre-rainfall leaf flush (Childes, 1988; Ryan et al., 2017).
In other words, anomalies in NDVI and hence woody vegetation cover, were anticipated to occur independently of anomalies in precipitation. Hence, from the un-coupled relationship observed between both variables, we may conclude that the model is, to a certain extent, effective at predicting woody cover, with the low $R^2$ supporting the effectiveness of using dry season phenological metrics to predicted woody cover. Finally, the high $R^2$ values observed throughout a portion of the western escarpment region may be a response specific to vegetation communities or species avoiding drought by not leafing-out in dry
years, such as the widespread *Senegalia reficiens* (pers. comm. C. van der Waal, 2017).



## 4.5 Regional hotspots

Namibia is comparatively heterogeneous in terms of eco-floristic regions and climate and consequently land-use, due to the pronounced altitudinal and climatic gradients (Mendelsohn and el Obeid, 2005c). Large parts of the south and western coast of the country are hyper-arid to arid and have very low woody vegetation cover, which may help explain why little or no

significant trends were identified here. In these regions the land is mainly used for extensive grazing with little or no cropping being practiced due to the extreme aridity (Mendelsohn et al., 2002). Figure 6, which illustrates significant trends in woody cover across Namibia, suggests that there are important differences in the spatial pattern of trends across the country; based on this observation, six areas were qualitatively selected for further discussion (Figures 5a to f).

The Kaokoland region (Figure 6a) exhibits important negative trends in woody cover; this region is primarily composed of

mopane woodland which are often used as coppice stands harvested for building material. Being sparsely populated during the civil war, the mopane woodland in this region is thought to have widely regenerated and may now be extensively utilized (Mendelsohn and el Obeid, 2005c).

The Ohangwena and Kavango West border regions (Figure 6b) display significant positive trends; these findings are somewhat unexpected, since  the area is known to be experiencing small-scale deforestation for urbanization and rain-fed crop farming

(Wingate et al., 2016). However, evidence suggests shrub encroachment is also occurring in these areas, potentially contributing to explaining the observed greening trends (De Klerk, 2004b; Erkkilä, A, 2001). Further, the moderate spatial resolution of MODIS is probably masking small-scale deforestation which is likely to be co-occurring adjacent to greening trends.

In the Kavango East region (Figure 6c), a spatially heterogeneous pattern of trends can be observed, with negative trends

predominating. Increasing fire frequency is the likely cause of this, as demonstrated by the long-term Namibian fire monitoring data (Anon, 2017). These data agree with a recent study demonstrating regional increasing fire activity (Andela et al., 2017).

Figure 6d highlights widespread positive trends; this area comprises both large and small-scale agriculture on communal land in the southeastern portion of the rectangle, as well as agriculture and tourism on freehold land on the north-western side. Despite the different land-uses present within the demarcated area, a very homogeneous trend is observed, possibly suggesting

widespread encroachment by *Senegalia mellifera* and *Dichrostachys cinerea* occurring across land-use types (De Klerk, 2004b).

Important negative trends in woody cover were identified across the region south of Etosha National Park (Figure 6e). These may be the result of several consecutive low rainfall years and a high density of elephants (*Loxodonta africana*), which in combination led to a net decline in woody cover (de Beer et al., 2006). The central eastern area (Figure 6f) which comprises

freehold land again reveals widespread positive trends, pointing to the occurrence of shrub encroachment by *Senegalia mellifera* (De Klerk, 2004b).



## 4.6    Trend analysis

Recent studies find a greening trend in satellite-derived vegetation proxies across southern Africa; however, interpreting these trends in terms of ecologically meaningful, measureable and identifiable vegetation properties, for instance, in terms of plant functional type, is often problematic (Brandt et al., 2015; Saha et al., 2015; Song et al., 2018; Zhu et al., 2016). We find woody cover has decreased overall and several recent studies corroborate these findings; for example, decreases in NPP were identified in the adjacent Okavango, Kwando and upper Zambezi catchment areas over a 29-year period using MODIS (Zhu and Southworth, 2013). Similarly, Andela et al. (2017) find an increasing trend in fire activity across much of the north east of the country which implies less woody cover (Andela et al., 2017; De Cauwer et al., 2016; Sankaran et al., 2008). These results stand in contrast to those of Tian et al. (2016) (Tian et al., 2016), who identify increases in woody density across southern Africa, and Fensholt et al. (2012) (Fensholt et al., 2012) who identify greening trends. However, both these studies rely on coarse spatial resolution data (i.e. 0.25°, 8 km and 0.05°× 0.05°, respectively) and cover different time periods (2000-201 and 1981-2007, respectively). Furthermore, they do not provide country-specific change statistics, but only regional approximations.

The south-north and west-east gradient of increasing percentage woody cover reflects the different eco-floristic regions (Figure 6). Northern regions consist of Kalahari and mopane woodlands, whereas the southern regions are made up of grass and shrub land, which exhibit lower woody cover densities (J Mendelsohn et al. 2002; John Mendelsohn and el Obeid 2005b). The overall decline in woody vegetation cover is therefore likely to be occurring in the north of the country, and be associated with declining tree and shrub stem numbers; these variables are related to aboveground biomass, as well as foliar and canopy density (Asner et al., 2003; Asner and Heidebrecht, 2005; Asner and Lobell, 2000). In effect, Wingate *et al.* (2018) identified a net loss of aboveground woody biomass for the northern Kalahari ecoregion (Wingate et al., 2018). Based on our results, we may conclude that decreases in vegetation biomass associated with woody vegetation are also taking place, especially in the desert and tropical dry forest biomes; however, since the approach used does not permit the direct estimation of change in carbon stocks, a precise inventory of loss and gains cannot be undertaken. Several anthropogenic and biophysical factors are known to drive decreases in woody cover. In particular, they include long-term changes in precipitation patterns, disturbances resulting from cattle grazing, high densities of browsers, fires, timber extraction and land clearing (De Cauwer et al., 2016; Sankaran et al., 2005, 2008).

Gains in woody cover are thought to be driven by factors including reforestation, conservation land management activities and raising atmospheric $CO_2$ concentrations, which under certain conditions have been found to favor C3 plants over C4, and the interaction of these factor presumably leading to shrub encroachment (Donohue et al., 2013; Mendelsohn and el Obeid, 2005a; Saha et al., 2015). Finally, most of the study area (92.2%) exhibits no significant trends, which agrees with several long-term studies demonstrating vegetation to be remarkably stable in the region (Buitenwerf et al., 2012; O'Connor et al., 2014; Rohde and Hoffman, 2012).



### 4.7    Model accuracy and limitations

Both predictor layers MaxWS and SINT are proxies for herbaceous vegetation; their low ranking is indicative of the higher correlation between field measurements of woody cover and phenological metrics characterizing woody vegetation. Similarly, when model predictions are compared to percentage tree cover, an increasing spread of values can be noted in the higher predicted percentage woody cover classes, as indicated by the black standard error bars (Figure 5) (Bastin et al., 2017). The low RMSE error observed suggests both datasets show a moderately good agreement, in spite of the fact that the datasets measure distinct variables using different methodologies, which makes their comparison prone to a multitude of confounding factors. Low $R^2$ values are the result of single outliers (percentage tree cover) within woody cover classes.

The low $R^2$ found when comparing observed and predicted values result from several limitations related to the phenological metrics, field measurements and nature of the savanna system. Limitations associated with the phenological metrics include, pixel resolution, spectral limitations, the use of monthly averages causing the lose of the  full suite of variation in NDVI values, and the temporal mismatch between the field observations and the coincident pixel phenological metric values (Mendelsohn and el Obeid, 2005a). Limitations associated with field measurements include, the different sample sizes (e.g. 100 point observations per site for the 2012 and 2014 datasets, compared to 160 point for the 2016 dataset); date of field data collection (i.e. 2012, 2014, 2016); and a modified methodology adopted for the samples collected in 2016. Moreover, the small field plot size may not be adequately representative at the spatial resolution of the MODIS data, and similarly, the field sample sites may not be sufficiently representative of the variability within predictor metrics (i.e. more variability within a pixel that between pixels) (Baccini et al., 2007). The seasonal phenological cycles of woody and herbaceous vegetation in response to precipitation and temperature cues, results in a variable NDVI signal for any given period. This causes the computed phenological metrics to vary annually, in relation to the static field plot estimates.  Furthermore, management actions, such as grazing and fire are likely to have impacted both the sample sites and the coincident pixel values of each metric.  In particular, fire scars, which are often extensive, cause the NDVI signal to fluctuate importantly and hence affect the phenological metrics. Fire is an important factor shaping vegetation structure and composition across Namibia, with certain areas routinely experiencing grass fires during the on-set of the dry season (John Mendelsohn and el Obeid 2005c). Lastly, the diverse vegetation characteristics, including species composition and structure, encompassed within the field sites used for model calibration, are likely to not be fully representative of the overall local to regional vegetation characteristics, resulting in decreased model accuracy (Carreiras, Vasconcelos, and Lucas 2012; V. R. Wingate et al. 2018). These may include for example, among other biotopes, wetlands and floodplains which are ephemerally submerged in water, and where herbaceous perennial plants can remain green for longer periods (Hüttich et al. 2011).

Several additional limitations are are likely to have introduced inaccuracies into the modelling of woody cover. For instance, NDVI is well correlated with vegetation chlorophyll content, leaf color, vegetation density and depth, soil color and moisture, as well as being a good indicator of NPP in drylands. However, it is limited by effects of soil and senesced vegetation background and signal saturation at high biomass levels, while in addition not being directly correlated with woody cover





(Asrar et al., 1984; Pettorelli et al., 2005; Prince, 1991; Sellers et al., 1992). Furthermore, savanna biomes are often characterized by several vegetation strata, ranging from tall tree canopies to shrub and herbaceous layers, all of which exhibit distinct phenophases (Chidumayo, 2001). Taken together, this variability in these factors contributes to impacting the regularity and rigorousness of the phenological metrics extracted.

## 5    Conclusion

This study provides a new estimate of change in woody cover across Namibia. Annual maps were created based on contemporary field measurements and MODIS NDVI metrics aimed at enhancing the distinct phenophases of woody and herbaceous vegetation. The resulting time-series was used to map trends in woody cover, which are excellent indicators of vegetation changes, including shrub encroachment and deforestation. The annual rate, trajectory and spatial extent of change

was evaluated in relation to potential drivers, including biomes, land-use, population density and precipitation.

On average, a loss of woody cover was identified; specifically, the desert and tropical dry forest biomes displayed a marked decline in woody cover, pointing to long-term land degradation, and deforestation/forest degradation, respectively. In contrast, tropical shrub lands demonstrated increases in woody cover, suggestive of woody encroachment. These results reflect those of a recent pan-African study on trends in woody cover (Brandt et al., 2017). Here, we identify contrasting change processes,

where woody cover loss is associated with more humid areas (tropical dry forest), and very arid areas (tropical desert), while woody cover gain predominated across the intervening tropical shrub lands.

Certain land-uses exhibited pronounced declines, notably protected areas; here these changes may be due to woody vegetation die-back caused by large herbivores and below average rainfall. Similarly, a negative trend was identified in resettlement and small-scale communal agricultural land, and is likely the result of increases in urbanization, deforestation and fire frequency;

similarly, a negative trend on freehold might be the result of encroacher shrub control. Greening trends across large-scale agriculture on communal land could be indicative of shrub encroachment, agro-forestry and fencing causing decreased grazing intensity. Importantly, no significant trends in woody cover were found across most of the country.

Qualitative high resolution image interpretation allowed the nature of observed land cover changes to be evaluated; in particular, our trend analysis effectively captured direct human impacts such as land clearing. However, greening could not be

conclusively identified using available imagery, and is probably the result of indirect impacts. Lastly, trends in woody cover and trends in precipitation are unrelated for most of the study area; their un-coupled relationship supports the validity of using metrics which enhance the distinct phenophases of woody and herbaceous vegetation.

Our results point to a landscape substantially affected by direct human impacts, resulting from the expansion of agriculture and urbanization, but also from indirect impacts, manifesting as long-term gradual vegetation changes. Moreover, distinct

change processes prevails across different biomes. Both instances have important implications for the provision of long-term ecosystem services, and evaluating the response of biomes with large proportion of C4 species to changing atmospheric $CO_2$ concentrations.



## Acknowledgments

The authors would like to thank Dr France Gerard for her efforts in the editing of this manuscript and providing valuable insight into the remote sensing aspects.

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
