# Peer review of "Mapping trends in woody cover throughout Namibian savanna with MODIS seasonal phenological metrics and field inventory data"

_Biogeosciences, 2019_

## Referee Comment (RC1) · Anonymous Referee #1 · 20 Mar 2019

I think this is an interesting topic but the current weak structure, somewhat sloppy writing and lacking information makes the reporting of the study weak, even if it may be OK when done. In general consider the following points for major revisions before it can be reconsidered.

* Don't write longer than needed. Keep it precise, clear and brief.

* Make sure the methods are described in such detail that an informed colleague can repeat the study based on the information provided in the met-section. This is not the case at the moment.

* Use SI units

[Figure]

* Better illustrations and captions needed. The reader should be able to get the context by looking at the figure and reading the caption, with no need to consult the bulk text. Examples: Figure 3. Explain abbreviations used in figures (DSINT etc.) Figure 4. Replace x and Y with real variables and units. Confidence interval for the regression line? Figure 5. Percentage tree cover range from 0-0.8% i.e. very low. Should be 0-80%? Figure 6. Add units to the colour bars (Slope%, woody cover %) Figure 7. Unclear content and message

* Let each section (Methodology, results etc.) contain information related to that section only, i.e. don't mix methods and background etc.

* Provide some justification for using NDVI and what it measure, Saturation effects due to higher LAI etc. Why not EVI or PPI?

Some additional comments in attached PDF

Please also note the supplement to this comment:
https://www.biogeosciences-discuss.net/bg-2019-28/bg-2019-28-RC1-supplement.pdf

―――――――――――――――――

---

## Author Comment (AC1) · 22 Mar 2019

Dear Sir/Madam, Please see the attached file supplement which contains a covering letter and addresses each reviewer comments. In addtion, the plain text if copy and pasted below. Kind regards, Vladimir Wingate

Natascha Töpfer Copernicus Publications Editorial Support editorial@copernicus.org

Handling Associate Editor: Anja Rammig, anja.rammig@tum.de

Journal: BG Title: Mapping trends in woody cover throughout Namibian savanna with MODIS seasonal phenological metrics and field inventory data Author(s): Vladimir R.

Wingate et al. MS No.: bg-2019-28 MS Type: Research article

Dear Ms Töper, dear Ms Rammig, We are grateful for the opportunity to resubmit a revised version of our paper "Mapping trends in woody cover throughout Namibian savannah with MODIS seasonal phenological metrics and field inventory data", and would like to thank you for your assistance in the submission process. We would also like to thank the anonymous reviewer for providing a very constructive review that has resulted in a strengthened manuscript. Please find attached a response to the reviewer comments, which provides a point-by-point response to each of the referee's comments, and where we have endeavoured to concisely address each point raised. The manuscript has undergone the Major and Minor Revisions suggested by the reviewer, including the re-structuring sections, the correction of figures and alteration of text. These changes are noted in the response file (below) together with the page and line number, and are highlighted in yellow. Since every review comment was highly appropriate and valuable, we have followed the recommendation of the reviewers as much as possible. I hope that you continue to find this research engaging and much look forward to hearing back from you in due course. Best regards,

Vladimir Wingate

On behalf of Prof Nikolaus Kuhn, Prof Stuart Phinn, Dr Cornelis van der Waal
I think this is an interesting topic but the current weak structure, somewhat sloppy writing and lacking information makes the reporting of the study weak, even if it may be OK when done. In general consider the following points for major revisions before it

can be reconsidered.

* Don't write longer than needed. Keep it precise, clear and brief.

As per the reviewers suggestions (minor revisions), certain sentences and words have now been omitted.

* Make sure the methods are described in such detail that an informed colleague can repeat the study based on the information provided in the met-section. This is not the case at the moment.

As per the reviewers recommendations (minor revisions), the Methods section has now been amended in order that the study may be repeated in full.

* Use SI units

Since the spatial resolution of Corona imagery was report in feet on the USGS Earth Explorer websites, the authors have chosen to retain the original unit. Please refer to this website: https://www.usgs.gov/centers/eros/science/usgs-eros-archive-declassified-data-declassified-satellite-imagery-1?qt-science_center_objects=0#qt-science_center_objects

* Better illustrations and captions needed. The reader should be able to get the context

by looking at the figure and reading the caption, with no need to consult the bulk text. Examples: Figure 3. Explain abbreviations used in figures (DSINT etc.) Figure 4. Replace x and Y with real variables and units. Confidence interval for the regression line? Figure 5. Percentage tree cover range from 0-0.8% i.e. very low. Should be 0-80%? Figure 6. Add units to the colour bars (Slope%, woody cover %) Figure 7. Unclear content and message.

Figure 3: captions have now been expanded as per the reviewers recommendation, and now reads: "Figure 3. Predictor variable importance (2008) generated using the Random Forest algorithm, evaluated using Mean Standard Error (MSE) (a) and node

purity (b): here, predictor variables associated with woody vegetation are consistently (i.e. DSINT) are consistently more importance that those associated with herbaceous vegetation (MaxWS). In addition, Mean coefficient of variation is mapped for the study area, and revealing greater uncertainty in arid coastal regions." Figure 4: x and y axes have now been replaced with real variables as per the reviewers suggestion, in addition, a confidence interval (0.95) has been included for the regresdsion line.

Figure 5: Percentage tree cover range has now been modified to 0-80% as per the reviewers recommendations.

Figure 6: Units and colour bars (Slope%, woody cover %) have now been added as per the reviewers recommendations and read: "Slope% km2 yr-1."

Figure 7: Unclear content and message:

As per the reviewers recommendations, the caption has been developed to better describe the content and message of Figures 7 and 8.

P20L1: The text now reads: "Figure 7. A qualitative assessment of what the observed trends represent on the ground, in terms of land cover changed, was undertaken by visual assessment of multi-temporal, high resolution imagery and random sampling. Here, a randomly sampled point for an area exhibiting a significant negative slope ($\geq$-25%) is presented and found to manifest as land clearing for small-scale agriculture and indicative of direct land cover change. These are identified using a 1972 Corona image (a) and a 2010 aerial othrophoto (b).".

P22L1: Figure 8. A qualitative assessment of what the observed trends represent on the ground, in terms of land cover changed, was undertaken by visual assessment of multi-temporal, high resolution imagery and random sampling. Here, a randomly sampled point for an area exhibiting a significant positive slope($\geq$25%) is presented and found to manifest as no apparent change that can be identified from a 1972 Corona image (a) and a 2010 aerial othrophoto (b). Results may be indicative of indirect change.

* Let each section (Methodology, results etc.) contain information related to that section only, i.e. don't mix methods and background etc.

As per the reviewers suggestion, certain paragraphs and sections found on the methods which give background information have been moved to the Introduction section, specifically, the following paragraph introducing trend analyses using EO data was moved to the introduction P2L18: "Key aspects surrounding trend estimation from Earth Observation (EO) time-series include temporal and spatial resolution, as well as data quality (Badreldin and Sanchez-Azofeifa, 2015; Sulkava et al., 2007). Although trend estimation using linear regression analysis is widely employed, it contravenes several statistical assumptions (deBeurs and Henebry, 2004; Eklundh and Olsson, 2003). Hence, non-parametric tests which overcome these limitations were applied (i.e. Mann-Kendall and Median Theil Sen trend analyses) (deBeurs and Henebry, 2004; Forkel et al., 2013). Furthermore, limitations are incurred by temporally aggregating, for example, to the annual scale, by diminishing temporal resolution. On the other hand, annual aggregation may strengthen trend analysis by eliminating seasonal cycles, which have been found to add seasonal correlation structures and thus augmenting uncertainties (Forkel et al., 2013).".

* Provide some justification for using NDVI and what it measure, Saturation effects due to higher LAI etc. Why not EVI or PPI?

As per the reviewers suggestions, the authors have elaborated on the justification for using NDVI and what it measures. Spefically: L5P3: This sentence now reads "These indicators are often derived from spectral vegetation indices of satellite imagery, which are related to the vegetation density of canopies". The issue of NDVI saturation is occurring at higher LAI is not applicable to this study, although this issue is discussed in L33P27. We now address why NDVI was used rather than EVI: P3L8: For this study, the authors have chosen to use NDVI rather than the Enhance Vegetation Index (EVI), since it has been shown to effectively capture vegetation density in savannah environments (Brandt et al., 2016a; Olsson et al., 2005; Wagenseil and Samimi, 2007).

Some additional comments in attached PDF

L18P2: scale is not suitable (1:1000 is a scale). Perhaps "extent"? This sentence has been changed as per the reviewer's recommendation. "Thus, there is an inadequate understanding of the extent of woody vegetation change in relation to environmental and socio-economic and environmental drivers". L27P2: Unclear. What is VHR? Spatial resolution? Temporal resolution? spectral resolution? Specify. This sentence now reads: "These products use very high spatial resolution scenes to train a vegetation cover algorithm based on high to moderate resolution imagery".

L5P3: What is photosynthetic potential? Define. What about NDVI saturation occurring at higher LAI? Can it be a problem here?

L5P3: This sentence now reads "These indicators are often derived from spectral vegetation indices of satellite imagery, which are related to the vegetation density of canopies". The issue of NDVI saturation is occurring at higher LAI is not applicable to this study, although this issue is discussed in L33P27.

L29P3: Ref to support this? This sentence now reads: "For example, a pre-rainfall leaf flush and synchronized flowering is commonly observed in three tree/shrub species which are widespread in the northeast, in particular, Terminalia sericea, Ochna pulchra and Pterocarpus angolensis (Childes, 1988)".

L1P4: Unclear which literature support these statements. The following references have now been included: "The annual growth of herbaceous biomass relies on the first precipitation events to initiate photosynthesis and remains photosynthetically active during the rainy season, as it is largely dependent on the spatio-temporal distribution of annual precipitation (Mendelsohn and el Obeid, 2005a). Senescence of herbaceous vegetation then takes place at the onset of the dry season once the plants have completed their annual life cycle, while in addition, intense grazing pressure throughout the country contributes to promptly grazing the pasture throughout much the country (Mendelsohn and el Obeid, 2005a). Importantly, this results in woody vegetation remaining photosynthetic during part of the year, while herbaceous vegetation is entirely desiccated (Verlinden and Laamanen, 2006b).”

1.3 Aims: Le this section include the aim only. Add the motivation above. As per the reviewrs suggestion, this section has been renamed to “Motivation and aims”.

L29P4: extent “Regional scales” is acknowledged by the authors to be accepted terminology in global ecological studies, and has therefore not been changed, please refer to this article: http://science.sciencemag.org/content/241/4873/1613

L32P4: reviewer deleted: “In addition, vegetation change processes, including deforestation and woody encroachment are reported to be widespread in Namibia, yet their spatial and temporal dynamics remain little studied.”. This sentence has now been removed as per the reviewer suggestion.

L6P8: Give enough details on the S-G filter that it can be repeated by another user. L8P8: This sentence now reads: “A Savitzky-Golay (SG) smoothing filter was then applied (using the default SG filter settings available in TIMESAT) to each pixel of the time-series to interpolate missing values, smooth outliers and minimize the effects of low quality data resulting from noise and cloud cover, and the time-series was aggregated to mean monthly values”. L6P8: THeera re some issues with the quality data according to https://lpdaac.usgs.gov/dataset_discovery/modis/modis_products_table/mod13q1_v006 Known Issues The following issues have been detected: Unexpected missing data in the last cycles of each year. Incorrect instances of "NoData" and spikes in NDVI values. VI Usefulness Bits are not correctly assigned. For instances where the VI Quality (bits 0-1) is flagged as good and the VI Usefulness (bits 2-5) indicates the same pixels have the lowest usefulness score, users are advised to disregard the usefulness score. Corrections will be implemented in Collection 6.1 reprocessing in 2019. Make sure the Quality data handling is clear enough so it can be repeated based on provided information. Provide some detail on the processing of the QI data. L7P8:

This issue has now been addressed: “Pixels flagged as low quality were masked; here, only values with a pixel reliability summary QA of 0 were used (where is equal to good data which can be used with confidence). Why was not EVI used/tested? It has frequently been shown to perform better than NDVI in semi arid regions. For this study, the authors chose to use NDVI, since it is still frequently used to monitor vegetation change globally using MODIS, in addition, it allow for comparison with previous studies.

L21P8: reviewer deleted “values” P8L19: This sentence now reads: The post-processing and sampling effort was also different for the 2016 dataset, in which data were processed to fractional cover.” L5P9: And how was this homogeneity assessed? Visual inpsection?. It is very hard to select a homogenous are for a 250x250 meter pixel when on the ground, at least if there are trees and shrubs in the area. Describe. This sentence now reads: “We justify this assumption since the field plots were sampled in homogenous vegetation strata (Baccini et al., 2007). Homogeneity was assessed via visual inspection of high resolution imagery and where possible extensive field observations of vegetation cover and composition”

L8P9: Give URL, ref and data set used. Valid for which time period? P9L8: This sentence now reads: “Biomes distribution was downloaded from the Food and Agricultural Organization Global Forest Resources Assessment (http://www.fao.org/3/ad652e/ad652e00.htm); for Namibia, they comprise tropical desert, tropical dry forest, tropical mountain system and tropical shrub land, the latter two being very similar (Simons et al., 2001).”

L10P9: Population density for which time period)s)? L9P10: This sentence now reads: “Population density data were obtained from the Worldpop, high resolution global gridded dataset at 100 m resolution, which gives an estimation of the number of people per km2 in 2015 (Lloyd et al., 2017)” L17P9: give resolution im as well. The homepage (https://climatedataguide.ucar.edu/climate-data/cmorph-cpc-morphing-technique-high-resolution-precipitation-60s-60n) says 0.25X0.25 deg.

L9P16: This sentence has been changed to: "Monthly precipitation was computed using the Climate Prediction Center Morphing technique (CMORPH) dataset, in which precipitation estimates are from satellite-derived passive microwave and infrared data, and available at a resolution of 0.25° (Joyce et al., 2004)."

L18P9: how? Reference period? (anomalies) L18P9: This sentence now reads: "The CMORPH dataset was aggregated to mean annual values and converted to anomalies, based on the overall mean of the time-series." L19P9: with data for the same year or lagged? P9L18: This sentence now reads: "To evaluate the correlation between rainfall and modelled woody cover, the CMORPH anomalies time-series was regressed, as the independent variable, against the time series of annual percentage woody cover anomalies (no time lag were used)".

L17P10: reviewer deleted Eklundh reference. L17P10: As per the reviewers suggestion the Jönsson and Eklundh et al has now been removed.

L25P10: reviewer deleted "two accuracy metrics, namely, the" L27P10: As per the reviewer suggestion, this sentence now reads: "The paired observed and predicted values were used to compute the Root Mean Squared Error (RMSE) and the coefficient of determination (R2) (Stehman et al., 2012; Willmott, 1982)

L3P11: And how many of these are located in Namibia? (sample plots Bastin) All sample plots are located in Namibia. This has now been specified in the text: L5P11: "Finally, model predictions were compared to the recently published 4,684 sample calibration/validation dataset of percentage tree cover from Bastin et al. (2017) (all plots located in Namibia)" L27P11: describe how this was done (how we converted to anomlaies) The anomalies calculated the deviation from the mean. P11L29: This has now been specified in the text: "The time-series was first converted to anomalies (deviation from the mean) before applying the trend analysis (Eastman, 2009).

L17P12: This need to be clarified. (1000*1000)/(250*250) = 16, not 1.6? Equation should read: 100, 0000, and the expansion factor should be 16. In consequences,

the necessary amendments have been made throughout the text and tables. L25P12: per year? This sentence has been changed and now clarifies that: P12L25: "Two classes were created representing areas mapped as either positive or negative trends, with slopes $\geq$25% ($\geq$-25%), using the final Theil-Sen slope image." In addition, the "%" in ($\geq$-25%) has now been included. L26P12: Please useSI units-Since the spatial resolution of Corona imagery was report in feet on the USGS Earth Explorer websites, the authors have chosen to retain the original unit. Please refer to this website: https://www.usgs.gov/centers/eros/science/usgs-eros-archive-declassified-data-declassified-satellite-imagery-1?qt-science_center_objects=0#qt-science_center_objects L1P13: this term is not mentioned before and not in the met section. What is it. Explain and define. Does it include MSE and Node purity? How? (predictor layer importance). Predictor variable importance evaluation is introduced in the Results section 2.9 Model accuracy and comparison: "Two measures are used to assess predictor variable importance, including percent increase in Mean Standard Error (MSE) following random permutation, and increase in node purity resulting from all the splits in the forest based on a particular variable, as computed using the gini criterion (please refer to Breiman, 2001 for details)." L6P13: What is the difference between variable importance and predictor importance? Explain. This sentence has now been changed to: "predictor variable importance" This sentence now reads: "Predictor variable importance (2008) is plotted in (Figure 3); two measures are used to assess predictor variable importance ..." L7P13: Explain and define. (gini) This sentence now reads: "...and increase in node purity resulting from all the splits in the forest based on a particular variable, as computed using the gini criterion (please refer to Breiman, 2001 for details)" Figure 3 P13: Label subfigs a,b,c so they can be identified. Labels in the sub-figure have now been changed as per the reviewers suggestions. Figure 4: observations >10% wooduyt cover was removed? Not visible here. Add 1:1 line and make the graph quadratic so This should read: observations >1% removed. P8L24: This sentence now reads: Samples with a measured percent woody cover <1% were excluded (n=25) from this analysis in order to apply log

transformations, which otherwise would have resulted in negative values, this resulted in a total of 458 available for model calibration. L4P14: Figure 4 reports and R2 of 0.467!? And RMSE of 14.47% This sentence now reads: "Figure 5 illustrates the linear relationship between percentage woody cover at 5% increment classes (2016), and percentage tree cover, yielding an R2 of 0.77 and an RMSE of 3.94%"

Figure 5: this is a very low percentage with max at 0.8% Should be 80%? Figure has now been updated and reads 80%

L6P15: How is this to be interpreted? [% km2 yr-1] Slope (percentage change in NDVI) per Km2 per year)

Please also note the supplement to this comment:
https://www.biogeosciences-discuss.net/bg-2019-28/bg-2019-28-AC1-supplement.pdf

---

## Referee Comment (RC2) · Anonymous Referee #2 · 24 Apr 2019

Review of "Mapping trends in woody cover throughout Namibian savanna with MODIS seasonal phenological metrics and field inventory data"

The study aims to map, describe and explain trends in woody vegetation cover in Namibia. In the current shape of the manuscript, I cannot recommend a publication in Biogeosciences.

The methods and results are not described/presented in a way that allows to understand the results. The methods need to provide more details about how phenological metrics are defined and how they were computed. In addition, the description of random forest setup need to be substantially improved. From the Methods section, I

assumed that one random forest model was trained for all field observations but then on page 13 line 4, a "2007 model" is stated. So did you train random forest models for each year separately? In addition, it is not clear which predictors were included in the random forest models. Table 1 lists DSI, MeanDS, MaxWS and DSINT but then other predictors (partly year-specific) are mentioned in Figure 3. Please replace Table 3 with an overview Table about all random forest models that were trained and which predictors were included in each model.

I find it also confusing that different results for different years are shown. For example, Figure 3 shows the importance of predictors for 2008 but Figure 4 shows the random forest fits (? - axis labels are missing!) for 2016. Is there any reason why you selected these different years or why you are not showing results for all years?

Relation to previous studies: The authors refer to Brandt et al. (2017) and Song et al. (2018). These two studies show for Namibia an increase in woody vegetation cover and in short vegetation, respectively, which is contrary to the results of this study. A more direct comparison and discussion of these results is necessary.

Specific comments

Section 2 "Material and methods": This section misses currently a logical structure because it jumps back and forth between data description, methods, different types of datasets etc. I suggest revising the structure as following:

The current section 2.2 "Study region" (including Fig. 1) can be easily merged with the section 1.2 of the introduction. Then the new structure could be:

2.1 Method overview (= 2.1 Approach + reference to Figure 2)

2.2 Datasets

2.2.1 Field data (= 2.4 + 2.5)

2.2.1 Satellite and ancillary data (= 2.3 + 2.6 + 2.7)

2.3 Estimation of phenological metrics - includes smoothing filter description from 2.3 and a substantially improved description of the phenological metrics (2.8) and how their were estimated)

2.4 Estimation of woody vegetation cover (= 2.8 without the description of phenological metrics + 2.9 + 2.11)

2.5 Trend analysis

Sections 3 and 4: Both section have very similar sub-sections (e.g. 3.3. and 4.1 Trends in relation to biomes). Hence, the entire text is very lengthy and repetitive. I suggest to combine sections 3 and 4 into "Results and discussions". Please also assess if there are four different sub-section on "Trends in relation to . . ." needed.

P 1 L 20-25: Please indicate for which periods trends were computed.

P 9 L 1-5: Please clarify if the average percentage woody cover is representative for the variability in a field plot.

P 9 section 2.8: This section describes both the random forest modelling approach and the phenological metrics. I suggest to split these into two sections because the description of phenological metrics is currently not understandable.

P 9 L 30 – P 10 L 3: I do not understand what you are trying to say here. Please revise the paragraph.

P 13 L 3-4: I would rather state first which variables are the most important ones.

All figures: Please remove the black background in all figures. This is a waste of ink if somebody one to print the paper.

Fig. 2: It is confusing that the legend ranges from +0.8 to -0.2; please reverse. The inset map of Africa hides parts of the data; please change.

Fig. 3: The names of the predictors do not correspond to the abbreviations listed in

Table 1. Make sure to list the correct names of ALL predictor variables in Table 1.

Fig. 5: The intention of this figure is unclear to me. What do you want to show here? What do the error bars show? Why are you comparing 2016 woody cover with tree cover (which year?).

Fig. 6: Their no units for the percentage changes.

Fig. 7 + Fig 8: These figures could be smaller and combined in one figure.

―――――――――――――――――――――

---

## Author Comment (AC2) · 9 May 2019

Review of "Mapping trends in woody cover throughout Namibian savanna with MODIS seasonal phenological metrics and field inventory data" The study aims to map, describe and explain trends in woody vegeta-

tion cover in Namibia. In the current shape of the manuscript, I cannot recommend a publication in Biogeosciences.

The methods and results are not described/presented in a way that allows to understand the results. The methods need to provide more details about how phenological metrics are defined and how they were computed. In addition, the description of random forest setup need to be substantially improved.

The authors have now substantially altered the methods and results as per the reviews recommendations.

Specifically, 1) sentences which describe how phenological metrics are defined and computed have now been made clearer; 2) a description of the random forest models and how they were used has been updated, and 3) the structure of the methods has been altered as proposed by the reviewer (please see below for details).

From the Methods section, I assumed that one random forest model was trained for all field observations but then on page 13 line 4, a "2007 model" is stated. So did you train random forest models for each year separately? In addition, it is not clear which predictors were included in the random forest models.

Random forest models were trained for each year using one single set of field data, and the five corresponding phenological metrics values for each year (2001-2016). This point has now been clarified as per the reviewer's suggestions.

L19P11 now reads: "Models were created by taking plot measurements of percent woody cover, with the coincident pixel values of each of the five metrics (Table 1), for every year (2001-2016)".

Table 1 lists DSI, MeanDS, MaxWS and DSINT but then other predictors (partly year-specific) are mentioned in Figure 3. Please replace Table 3 with an overview Table about all random forest models that were trained and which predictors were included in each model.

Table 1 includes a description of each of the phenological metrics used.

All predictors where used in training the models.

L18P11 now reads: "Models were created by taking plot measurements of percent woody cover, with the coincident pixel values of each of the five metrics (Table 1), for every year (2001-2016)."

Rather than include a large table with all the individual predictor layers, we now indicate that the year is appended to the abbreviated form.

Table 1 now reads: "Table 1. Phenological metrics used in this study, their abbreviation and concise description. Each metric was computed for every years of the study (2001-2016); the resulting short form is then labelled as "DSI2016", where the year is appended to the short form."

I find it also confusing that different results for different years are shown. For example, Figure 3 shows the importance of predictors for 2008 but Figure 4 shows the random forest fits (? - axis labels are missing!) for 2016. Is there any reason why you selected these different years or why you are not showing results for all years?

Only the most recent model outputs are now included. The remaining model output figure are included in the supplementary material, since including them all would take too much place, and results in a very condensed figure. Instead, the text describing these results is included.

L6P14 now reads: "Predictor variable importance for the most recent output (2016) is plotted in (Figure 3). Plots for the remaining years (2001-2015) are provided in the supplementary material (Figure S1)."

Axis labels have now been included in Figure 4, and a 95% confidence interval has been added.

Relation to previous studies: The authors refer to Brandt et al. (2017) and Song et al.

(2018). These two studies show for Namibia an increase in woody vegetation cover and in short vegetation, respectively, which is contrary to the results of this study. A more direct comparison and discussion of these results is necessary.

We directly compare and evaluate the results from Song et al. (2018) for each biome in this study: Section 4.1.1, L15P23, as per the reviews recommendation. L15P23, now reads: "When evaluating the results from Song et al. (2018) for Namibia only, an overall greening trend from 1982 to 2016 can be identified. On average for each FAO biome, a decrease in bare ground and a simultaneous increase and short vegetation can be noted, while a gain in tree canopy is seen across the tropical dry forest biome."

Both Brandt 2017 and Song et al. 2018 use different datasets to evaluate changes in vegetation cover: specifically, these datasets a of a longer temporal resolution, and coarser spatial resolution, as well as different spectral resoltions. These specifics are now discussed in detail in Results Section 4.1: In addition, the results of Song et al. 2018 and Brandt et al. 2017 are compared to our results in the Results Section 4.1.

Specifically, with Reference to Brandt et al. (2017): Since the results from Brandt et al. (2017) show only a greening trend across Namibia, this study only attempts to discuss the potential causes of the differing results, and goes on to describe how they are similar across biomes.

Lastly, a sentence describing the different datasets used by Brandt et al. (2017) has been appended:

L7P24 now reads: "However, in contrast to the results of this study, they find a greening trend predominating across Namibia; this discrepancy can most likely be explained by the different temporal, spatial and spectral resolutions of the datasets used. Here, Brandt et al. (2017) employed the 0.25° spatial resolution 1992-2011 vegetation optical density dataset, derived from satellite passive microwave measurements."

Specific comments

Section 2 "Material and methods": This section misses currently a logical structure because it jumps back and forth between data description, methods, different types of datasets etc. I suggest revising the structure as following:

The current section 2.2 "Study region" (including Fig. 1) can be easily merged with the section 1.2 of the introduction. Then the new structure could be: Section 2.2 has now been merged with Section 1.2, and is now entitled Section 1.3 study region. 2.1 Method overview (= 2.1 Approach + reference to Figure 2)

Section 2.1 is now equal to 2.1 Approach, and a reference to Figure 2 has been included.

2.2 Datasets 2.2.1 Field data (= 2.4 + 2.5)

2.2.1 Satellite and ancillary data (= 2.3 + 2.6 + 2.7) The structure of the methods has now been adjusted as per the reviewers recommendations: 2.3 Estimation of phenological metrics - includes smoothing filter description from 2.3 and a substantially improved description of the phenological metrics (2.8) and how their were estimated)

Smoothing filter description was carried out as part of the initial MODIS processing, as per standard protocol (please see "TIMESAT—a program for analyzing time-series of satellite sensor data"); hence, the authors have decided to leave the smoothing paragraph in its current section.

A description of the phenological metrics has now been enhanced as per the reviewer's suggestions (Section 2.3).

Specifically, this section clarifies that these metrics were computed annually.

For a complete description of how certain of the metrics were computed, namely, the DSI, DSINT, and SINT, the authors continue refer the reviewer and readers to the relevant literature, in order to keep the length of the text to a minimum. For the remaining metrics, their estimation is included on the text.

2.4 Estimation of woody vegetation cover (= 2.8 without the description of phenological metrics + 2.9 + 2.11)

These sections have now been altered as per the reviewers suggestions.

2.5 Trend analysis Sections 3 and 4: Both section have very similar sub-sections (e.g. 3.3. and 4.1 Trends in relation to biomes). Hence, the entire text is very lengthy and repetitive. I suggest to combine sections 3 and 4 into "Results and discussions". Please also assess if there are four different sub-section on "Trends in relation to : : :" needed.

This manuscript had originally been in the "Results and Discussion" format, however, an earlier reviewer suggested that this be changed to two distinct Results and Discussion headings. As a consequence, the authors have decided to keep the present format, in light of the effort which went into separating these headings previously.

In addition, to make the text more comprehensible, a single section "4.1 Trend evaluation", now includes sections 4.1.1 Biomes, 4.1.2 Land-use and population, 4.1.3 Multi-temporal imagery assessment, and 4.1.4 Precipitation. These have been renamed so as to remove any repetition.

Finally, Results Section 3.3 has been renamed to "3.3 Trend evaluation in relation to biomes, land-use and population", in order to correspond more closely with heading 4.1

P 1 L 20-25: Please indicate for which periods trends were computed.

L23P1: The time period has now been included (2001-2016)

This section now reads: "An overall decrease in woody cover was identified over the period from 2001-2016, with the most pronounced decreases found in urban and densely populated areas."

P 9 L 1-5: Please clarify if the average percentage woody cover is representative for the variability in a field plot.

A single value of percentage woody cover was derived per plot using the methods proposed by Herrick et al., 2013. The within field plot variability was not available for this study, only percentage woody cover per plot.

P8L17 now reads: "Here, the woody cover per plot represents the % of points covered by either trees or shrubs or both, where each site/plot had 100-160 points."

In addition, P9L12 reads: "In this study, the field plots were not scaled to the resolution of MODIS (250 × 250 m) using spatial averaging; instead, instead, the percentage woody cover computed using the methods proposed by Herrick et al. (2013) within the 50 × 50 m field plot was compared with the corresponding MODIS pixel values"

P 9 section 2.8: This section describes both the random forest modelling approach and the phenological metrics. I suggest to split these into two sections because the description of phenological metrics is currently not understandable.

This section has now been split as per the reviewer's suggestion and are now refered to as Sections 2.4, 2.4.1, 2.4.2).

P 9 L 30 – P 10 L 3: I do not understand what you are trying to say here. Please revise the paragraph.

This paragraph has been revised and shortened and now reads: "

L15P11: "The Random Forest algorithm was selected since it is effective at estimating predictor variable importance, integrating multiple predictors variables with different predictive power, and not assuming normal statistical data distribution or any particular relation (i.e. exponential) between dependent and independent variables. Hence it has been used extensively in remote sensing studies to integrate a range of imagery and metrics (Breiman 2001; Moisen and Frescino 2002; Cutler et al. 2007; Prasad, Iverson, and Liaw 2006; V. Wingate et al. 2016)."

P 13 L 3-4: I would rather state first which variables are the most important ones. All figures: Please remove the black background in all figures. This is a waste of ink if

somebody one to print the paper.

A statement of which variables are the most important has now been included as suggested by the reviewer:

L3P14: This section now reads: "The evaluation of predictor variable importance yielded a clear pattern: the most important predictors were most often (but not in all cases) the mean dry season values (MeanDS), dry season index (DSI) and dry season integral (DSINT) (in variable order). The least important predictors were consistently the maximum annual wet season value (MaxWS) and annual small seasonal integral (SINT) (expect for the 2007 model, in which the DSINT is the weakest, potentially implying an anomalous year)."

Black background has now been removed from all figures as per the reviewer recommendation.

Fig. 2: It is confusing that the legend ranges from +0.8 to -0.2; please reverse. The inset map of Africa hides parts of the data; please change.

Figure 2: The legend has now been modified as per the reviewer recommendation: High NDVI values are shown in dark green, while low and negative values shown in beige. In addition, the inset map has been modified as per the reviewers suggestion.

Fig. 3: The names of the predictors do not correspond to the abbreviations listed in Table 1. Make sure to list the correct names of ALL predictor variables in Table 1.

A sentence clarifying that the phenological metrics (predictor variables) were computed for each year has now been included in the main text and within the table caption. Table 1 now reads: "Table 1. Phenological metrics used in this study, their abbreviation and concise description. Each metric was computed for every years of the study (2001-2016); the resulting short form is then labelled as "DSI2016", where the year is appended to the short form."

L21P10 now reads: "Phenology metrics for each year (2001-2016) were extracted using TIMESAT software, which has been extensively used for measuring seasonal land surface phenology in drylands"

Fig. 5: The intention of this figure is unclear to me. What do you want to show here? What do the error bars show? Why are you comparing 2016 woody cover with tree cover (which year?).

The aim of this section is to use an independent proxy variable (tree cover) and compare it to woody cover generated in this study, in order compute validation statistics.

Firstly, we compare tree cover and woody cover, measured as part of this study, and find they are closely correlated.

Subsequently, we use tree cover from the Bastin et al. (2017) dataset and compare it to the woody cover dataset generate in this study, and find a strong linear correlation. This correlation we use a validation statistic.

L6P12: This section now reads: "Finally, model predictions were compared to the recently published 4,684 sample calibration/validation dataset of percentage tree cover from Bastin et al. (2017) (all plots located in Namibia) (Bastin et al. 2017); each data point consists of a 0.5-ha plot, visually assessed for tree cover percentage using very high resolution imagery. Since we find observed tree cover percentage and observed woody cover percentage, sampled as part of this study, to be highly correlated (r=0.83), we assume the Bastin et al. (2017) dataset is assumed to act as a good proxy for percentage woody cover; it provides the latest estimate of tree cover in drylands and is based on the interpretation of contemporary high resolution imagery."

The caption of Figure 5 now reads: "Linear relationship between percentage woody cover (2016) at 5% increment classes, and percentage tree cover. Here, each annual estimate of percentage woody cover was compared to percentage tree cover derived from the Bastin et al. 2017 dataset. Errors bars show the standard deviation.."

Fig. 6: Their no units for the percentage changes.

Figure 6: Units have now been included

Fig. 7 + Fig 8: These figures could be smaller and combined in one figure.

Figures 7 and 8 have now been combined into a single figure as per the reviewers recommendation. Accordingly, the caption text has been revised.

Please also note the supplement to this comment:
https://www.biogeosciences-discuss.net/bg-2019-28/bg-2019-28-AC2-supplement.zip